# Interacting rhythms enhance sensitivity of target detection in a fronto-parietal computational model of visual attention

Amélie Aussel[1,2]*, Ian C Fiebelkorn[3,4], Sabine Kastner[4,5], Nancy J Kopell[1,2]*†, Benjamin Rafael Pittman-Polletta[1,2]*†

[1]Cognitive Rhythms Collaborative, Boston University, Boston, United States; [2]Department of Mathematics and Statistics, Boston University, Rochester, United States; [3]Department of Neuroscience and Del Monte Institute for Neuroscience, University of Rochester Medical Center, University of Rochester, Rochester, United States; [4]Princeton Neuroscience Institute, Princeton University, Princeton, United States; [5]Department of Psychology, Princeton University, Princeton, United States

*For correspondence: amelie.aussel@inria.fr (AA); nk@bu.edu (NJK); benpolletta@gmail.com (BRP-P)

†These authors contributed equally to this work

Competing interest: The authors declare that no competing interests exist.

**Abstract** Even during sustained attention, enhanced processing of attended stimuli waxes and wanes rhythmically, with periods of enhanced and relatively diminished visual processing (and subsequent target detection) alternating at 4 or 8 Hz in a sustained visual attention task. These alternating attentional states occur alongside alternating dynamical states, in which lateral intraparietal cortex (LIP), the frontal eye field (FEF), and the mediodorsal pulvinar (mdPul) exhibit different activity and functional connectivity at α, β, and γ frequencies—rhythms associated with visual processing, working memory, and motor suppression. To assess whether and how these multiple interacting rhythms contribute to periodicity in attention, we propose a detailed computational model of FEF and LIP. When driven by θ-rhythmic inputs simulating experimentally-observed mdPul activity, this model reproduced the rhythmic dynamics and behavioral consequences of observed attentional states, revealing that the frequencies and mechanisms of the observed rhythms allow for peak sensitivity in visual target detection while maintaining functional flexibility.

## Editor's evaluation

This valuable work introduces a detailed computational model to elucidate the underpinnings of experimentally observed coordinated rhythmic dynamics across brain regions. It provides a solid step towards understanding rhythmic attention. The work will be of interest to neuroscientists working on brain rhythms and attention from a cognitive, systems or computational perspective.

## Introduction

The term 'selective attention' refers broadly to a set of neural mechanisms that are tailored to boost the processing of behaviorally relevant stimuli, allocating the brain's limited processing capacity in an efficient way. Classically, it has been assumed that attention-related modulation of sensory processing is continuous over time during attentional deployment. However, there is now mounting evidence that, even during sustained attention, the attentional enhancement of visual processing waxes and wanes over time (*Benedetto et al., 2020*; *Fiebelkorn and Kastner, 2019b*). In a sustained visual attention task, periods of enhanced sensory processing at an attended location alternate with periods

of relatively diminished sensory processing, at a rate of 4–8 cycles per second (i.e. in the θ frequency range, *Fiebelkorn and Kastner, 2019b*).

This rhythmic sampling of the visual environment is apparent at the behavioral level: measuring the success of target detection as a function of the interval between presentation of a spatially predictive cue and a target reveals an oscillation in hit rates with 8 Hz periodicity at the cued location and 4 Hz periodicity at an uncued location (*Fiebelkorn et al., 2018*; *Fiebelkorn et al., 2013*; *Landau and Fries, 2012*; *Song et al., 2014*; *VanRullen et al., 2007*). Monkey and human electrophysiology has shown that this rhythmicity in spatial attention is reflected in the neural activity of various nodes of the attention network (*Fiebelkorn et al., 2018*; *Fiebelkorn et al., 2019c*; *Gaillard et al., 2020*; *Helfrich et al., 2018*). Recordings from the frontal eye field (FEF) and lateral intraparietal area (LIP) in monkeys have revealed that θ-rhythmic changes in behavioral performance are associated with θ phase-dependent dynamical states, which alternate at 4 Hz and are characterized by distinct neural dynamics and distinct patterns of functional connectivity between these areas (*Fiebelkorn et al., 2018*; *Fiebelkorn et al., 2019c*). These alternating states are also characterized by differences in thalamo-cortical interactions, with the mediodorsal pulvinar (mdPul)—a higher order thalamic nucleus associated with cognitive functions including attention (*Halassa and Kastner, 2017*; *Fiebelkorn and Kastner, 2019a*; *Kastner et al., 2020*; *Saalmann and Kastner, 2011*) and interconnected with the cortical nodes of the attention network (including LIP and FEF) in monkeys (*Gutierrez et al., 2000*; *Romanski et al., 1997*; *Selemon and Goldman-Rakic, 1988*)—specifically coordinating cortical activity during the θ phase associated with enhanced perceptual sensitivity (the 'good θ phase'), but not during the θ phase associated with relatively diminished perceptual sensitivity (the 'poor θ phase'; *Fiebelkorn et al., 2019c*).

The differences in circuit dynamics and functional connectivity observed during the good and poor θ phases suggest that, even during sustained attentional deployment, the attention system alternates between two states subserving different functional roles. Besides directing attention-related boosts in sensory processing, the attention network also coordinates orienting movements of the body (e.g. head movements) and eyes (e.g. saccadic eye movements) (*Corbetta et al., 1998*; *Moore and Fallah, 2001*). *Fiebelkorn and Kastner, 2019b* have proposed that rhythmic sampling reflects the temporal coordination of these sensory and motor functions, helping to avoid potential functional conflicts. That is, periodic dips in perceptual sensitivity (i.e. during the poor θ phase) may be associated with windows of opportunity for the motor system, during which it is easier to shift attention from the presently attended location to another location. Rhythmic sampling may thus provide critical cognitive flexibility, preventing excessive attention to any single stimulus in the environment.

Despite the appeal of this hypothesis, the neuronal circuit mechanisms that underlie the alternating dynamical and behavioral states observed during sustained attention remain unknown—as does whether and how their mechanisms contribute to the suppression and enhancement of sensory processing and motor execution. While the good θ phase is characterized by a parietal-driven increase in γ-band (~30–90 Hz) activity and a frontal-driven increase in $\beta_2$-band (~20–30 Hz) activity, the poor θ phase is characterized by increased parietal $\beta_1$-band (~12–20 Hz) activity (*Fiebelkorn et al., 2018*; *Fiebelkorn et al., 2019c*). Notably, γ-band activity has often been associated with enhanced sensory processing (*Fries, 2009*), and $\beta_2$-band activity has often been associated with the suppression of motor processing (*Gregoriou et al., 2012*; *Pogosyan et al., 2009*; *Zhang et al., 2008*). Consistent with these functional interpretations, during rhythmic sampling, γ-band activity is linked to LIP visual neurons (i.e. neurons that respond to visual stimulation but demonstrate no saccade-related activity), while $\beta_2$-band activity is linked to FEF visual-movement neurons (i.e. neurons that both respond to visual stimulation and demonstrate saccade-related activity; *Fiebelkorn et al., 2018*). Circuit mechanisms associated with these rhythms have been implicated in sensory processing, attention, and working memory (*Kramer et al., 2008*; *Gelastopoulos et al., 2019*; *Lee and Whittington, 2013*), but whether these mechanisms play a role in the complex rhythmic dynamics of the visual attention network is not known. In this study, we sought to leverage detailed information about neural dynamics during rhythmic sampling as well as insights from previous computational models to address the following questions: What are the mechanisms that generate the $\beta_1$, $\beta_2$, and γ oscillations observed in LIP and FEF, and how do these mechanisms and oscillations interact? How is this oscillatory activity flexibly controlled to vary with the θ rhythm, and how does it contribute to function? In particular, can FEF and LIP oscillations explain the variation in behavioral performance over the θ cycle? Finally,

are the specific frequencies observed in FEF and LIP, and the θ-frequency timing of these alternating attentional states, functionally significant?

To address these questions, we built and tested a computational model of FEF and LIP, representing the temporal dynamics of visual processing at the cued location following the presentation of an attention-priming spatial cue (leaving the allocation of attention among multiple objects and locations for future work). In our model, FEF and LIP were both driven θ-rhythmically by simulated mdPul input at α frequency. This is consistent with the experimental observation that mdPul exerts a Granger causal influence over FEF and LIP at an α frequency during a part of each θ cycle (i.e. at a particular phase of the θ rhythm; *Fiebelkorn et al., 2019c*). The model was constructed from biologically detailed neurons and is consistent with the known anatomy, physiology, and function of frontal and parietal circuits. It highlights the cell types and connections necessary to produce the dynamics of interest, to allow realistic insights into the mechanisms underlying these dynamics and their functional consequences, while also avoiding excessive complexity. That is, while other cell types and connections, besides those modeled, are present in the brain and may play important roles in other tasks and other aspects of this task, the present model suggests that the activity of these cell types plays a minimal role in the phenomena of interest (see Discussion, Modeling Choices & Limitations, and *Appendix 4—figures 1–3*). Since small models with single-compartment neurons can be difficult to parametrize using electrophysiological measurements, we built on previous cortical models that reproduced experimentally observed network dynamics in other cognitive and physiological contexts—specifically, in the contexts of working memory and cholinergic neuromodulation (*Kramer et al., 2008*; *Lee and Whittington, 2013*; *Gelastopoulos et al., 2019*). Incorporating these models as modules in our network increases the realism and explanatory power of our model, by further constraining its dynamics and function to match (qualitative) experimental observations.

Our model shows how θ-rhythmic input from mdPul to cortex, subsequent to a spatially informative cue, can produce alternating dynamical regimes that, in turn, explain periodicity in behavioral outcomes (i.e. hit rates). In a simulated sustained attention task, the model replicated the $\beta_1$, $\beta_2$, and γ dynamics observed experimentally. It offered an explanation for the frequency-specific causal interactions between LIP and FEF, suggesting a progression during enhanced stimulus processing in which pulvinar input activates FEF and prepares LIP to respond, but top-down $\beta_2$ input from FEF is necessary to elicit a γ-rhythmic response from LIP. The model also replicated behavioral results, unexpectedly revealing how an 8 Hz behavioral oscillation (at the cued location) can arise from a

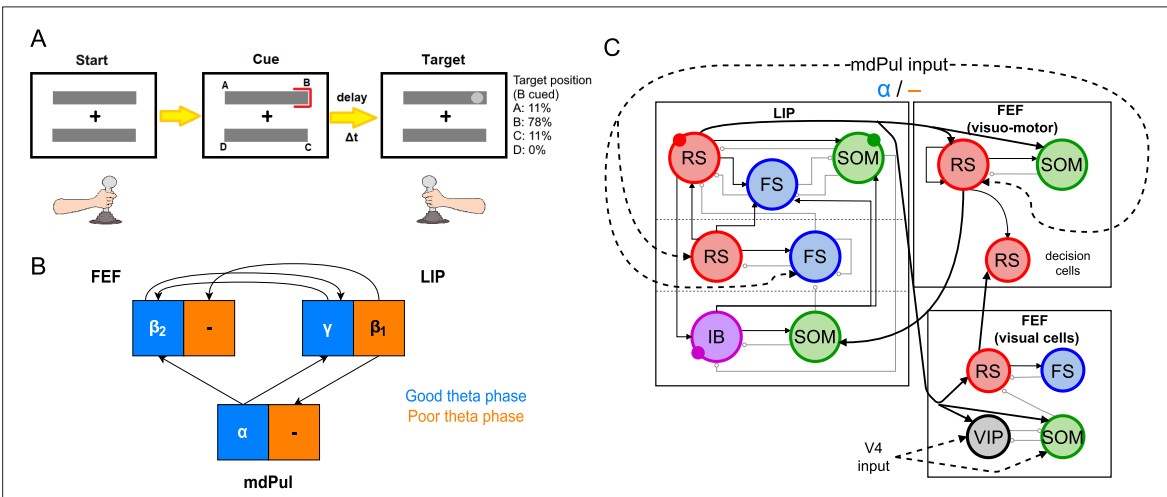

**Figure 1.** Overview of the Egly-Driver task, of the experimentally observed rhythms, and of our computational model.
(**A**). Schematic of the Egly-driver task modeled. (**B**). Neuronal rhythms experimentally measured in FEF, LIP and mdPul during the task. (**C**). Diagram of the full LIP and FEF model. Black lines ending in arrows and gray lines ending in circles indicate excitatory and inhibitory (chemical) synapses, respectively. Thick lines indicate synapses between modules. Dashed lines indicate synthetic inputs from regions that are not modeled. Circles indicate gap junctions between population neurons. The mdPul input is at an α frequency during the good θ phase and is absent in the poor θ phase. RS: regular spiking pyramidal neurons. IB: intrinsic bursting pyramidal neurons. FS: fast-spiking (e.g. parvalbumin-positive) inhibitory interneurons. SOM: somatostatin-positive interneurons. VIP: vasoactive intestinal peptide-positive interneurons.

4 Hz alternation of dynamical states, in dependence on the strength of functional connectivity from LIP to FEF. We found that altering the timescale of the oscillatory dynamics in FEF or LIP resulted in decreased task performance, providing evidence for the functional significance of these rhythmic dynamics. Finally, we found that model behavior was robust to some changes in model parameters, yet produced qualitatively distinct dynamics—characterized by different rhythms—in response to other parameter changes, which we hypothesize may represent other functional states of FEF and LIP. Thus, our results demonstrate the computational advantages of biologically realistic rhythmic network dynamics, suggesting that they enable visual attention networks to participate in attentional, motor, and cognitive tasks with sensitivity, flexibility, and versatility.

## Results

We modeled LIP and FEF activity during a sustained attention task based on the Egly-Driver task (*Fiebelkorn et al., 2019c*, see *Figure 1A*). In this task, the participant must respond (releasing a lever or pressing and holding down a button) when they detect a near-threshold contrast change in a visual scene. A trial begins when the participant fixates on the center of a screen (holding down a lever if one is used), on which appear two bar-shaped stimuli, presented vertically or horizontally with equal probability. After a short delay, a visual cue appears at the end of one of the bars. The target has a 78% chance of appearing at the same location as the cue, and an 11% chance of appearing at one of the two bar ends adjacent to the cued location (locations A and C in *Figure 1A*). After another delay of random duration lasting between 0.3 s and 1.6 s, the target appears in the form of a near-threshold change in contrast at the end of one of the bars, lasting 0.1 s. If the participant successfully detects the target, they then release the lever or press a button to receive a reward. Hit rates for this task can be analyzed as a function of the time of target presentation (relative to cue presentation), and were found in both humans and monkeys to exhibit θ-band periodicity, with alternating 'good' and 'poor' θ phases associated with increased and decreased hit rates, respectively, appearing at 4 Hz for the uncued object and 8 Hz for the cued object (*Fiebelkorn et al., 2013*; *Fiebelkorn et al., 2019c*).

Recordings from three of the brain regions involved in the task (the FEF, LIP, and mdPul) showed distinct patterns of activity between the good and poor θ phases (defined here based on behavioral performance at the cued location; *Figure 1B*). During the good θ phase (coded in blue in *Figure 1B* and throughout), power spectra, Granger causality, spike-field coherence, and cross-frequency coupling indicated that: mdPul provided α-rhythmic (10–15 Hz) drive to FEF and LIP; FEF generated $\beta_2$ oscillations (20–30 Hz) that drove LIP; and LIP generated γ oscillations (>30 Hz) that drove FEF. During the poor θ phase (coded in orange in *Figure 1B* and throughout), LIP produced dynamics at $\beta_1$ frequencies (~ 10–15 Hz), and this $\beta_1$ activity was Granger causal to activity in mdPul and FEF, even though these regions did not exhibit robust activity at any specific frequency (*Fiebelkorn et al., 2019c*).

*Figure 1C* shows the basic anatomy of the model. For LIP, we used a laminar model motivated by previous work on attention and memory (*Kramer et al., 2008*; *Lee and Whittington, 2013*; *Gelas-topoulos et al., 2019*). The model's dynamic repertoire of $\beta_1$, $\beta_2$, and γ rhythms depends on lamina-specific physiological properties and their interaction or isolation: superficial and deep layers can coordinate to produce a $\beta_1$ rhythm, while an input layer generates a γ rhythm and propagates it to the superficial layers, and deep layers resonate to $\beta_2$-frequency input. In FEF, there are (at least) three different functional classes of pyramidal cells, typically classified based on their sensory- and saccade-related response profiles. Visual neurons exhibit sensory-related responses, movement neurons exhibit saccade-related responses, and visual-movement neurons exhibit both sensory and saccade-related responses (*Fiebelkorn et al., 2019c*). In the experimental data we model here, too few FEF motor neurons were identified to draw reliable conclusions about their dynamics during the task (*Fiebelkorn et al., 2018*; *Fiebelkorn et al., 2019c*). Thus, we modeled FEF with two different functional modules: a visual module and a visuomotor module. We remain agnostic about the relationships of these functional modules to the laminar structure of FEF, since the geometry of FEF complicates laminar recording (*Bruce et al., 1985b*; *Selvanayagam et al., 2019*) and, thus, the laminar properties of FEF rhythms are not clear (although it is known that there are greater numbers of sensory and motor neurons in superficial and deep layers of FEF, respectively; *Pouget et al., 2009*).

All cell types were modeled as Hodgkin-Huxley point neurons, with spiking (i.e. leak, sodium, and potassium) currents tailored with cell type-specific parameters (see *Tables 1–5* in Methods).

In addition to two types of pyramidal cells (regular-spiking (RS) and deep intrinsic bursting (IB)), we modeled three types of interneurons. Parvalbumin positive interneurons were modeled as fast spiking (FS), and produced fast decaying inhibition (with a timescale of $5ms$), capable of pacing γ oscillations in networks consisting of interacting FS and RS cells. Somatostatin positive (SOM) interneurons contained an h-current in addition to the spiking current, and produced slower decaying inhibition (with a timescale of $20ms$; *Silberberg and Markram, 2007*), leading to $\beta_2$ oscillations when interacting with RS cells. Finally, vasoactive intestinal peptide (VIP) interneurons were modeled with a potassium D-current, and produced inhibition with a similar decay time to SOM cells, but spiked in response to much lower levels of excitation (*Appendix 1—figure 1*), in accordance with experimental data (*Tremblay et al., 2016*). VIP inhibition exclusively targets SOM interneurons in our model (see Methods).

All neurons in the model received tonic excitation at cell-type-specific levels, encapsulating the effects of cell-type-specific resting membrane potentials and/or background inputs (that do not vary with θ phase) from other brain regions. Each cell also received some random, gaussian-like excitation, and was initiated with a random membrane potential close to its resting potential (see *Table 6*). Certain cell types also received synthetic input, designed to match mdPul drive during the cue-target interval. This drive consisted of 12 Hz input nested in a 4 Hz θ rhythm; we assumed the θ rhythm would be sinusoidal or sawtooth-shaped, and that mdPul input would occur during the first half of the θ cycle (as seems to be the case experimentally; *Fiebelkorn et al., 2018*), so the second pulse of 12 Hz input was twice as large as the first. When isolated modules were analyzed, we substituted synthetic inputs for the inputs that would arise from other modules, based on the experimentally observed dynamics of those modules. (Synthetic inputs from LIP and FEF exhibited zero phase lag with the θ rhythm and mdPul input—that is, all rhythms were ongoing for the entire good or poor θ phase. Thus, simulated dynamics could occur at slightly different θ phases in isolated modules—as in *Figure 3* & *Figure 4*—vs. the full network—as in *Figure 2*.)

The rest of the Results section is organized as follows. In Mechanisms of FEF and LIP Oscillations During the Cue-Target Interval, we discuss model behavior during the cue-target interval and show that, when driven by θ-rhythmic inputs from mdPul and V4, the model reproduces the rhythms experimentally observed in LIP and FEF during both good and poor θ phases. In Mechanisms of Target Detection, we discuss model behavior on target presentation and show how the FEF model enables enhanced sensitivity and discrimination in target detection. In Task Performance, we share results on the functional implications of the model, and show the importance of the rhythms it generates. Finally, in Model Robustness and Flexibility, we discuss how parameters affect the model behavior, allowing it to exhibit different rhythms and possibly different functional states.

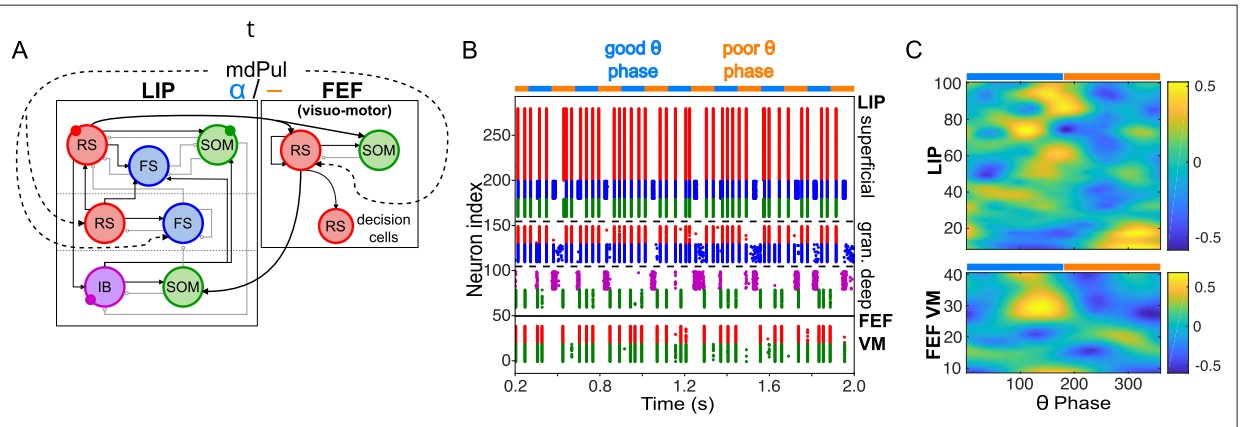

**Figure 2.** Model activity during the cue-target interval. (**A**). Diagram of the interconnected LIP and FEF visuomotor modules. Symbols are as in *Figure 1C*. (**B**). The LIP and FEF visuomotor modules reproduce experimentally observed rhythms when driven by mdPul. Raster plot from a 2 s long simulation with an alternation of good and poor θ phase pulvinar inputs. The activity of LIP is shown on top, and that of the FEF visuomotor module at the bottom. No target was presented, so decision-cells were quiescent and are omitted. C. Spectral power as a function of θ phase for simulation shown in **B**. LIP produces an alternation of γ and $\beta_1$ rhythms. FEF visuomotor cells produce $\beta_2$ oscillations only during the good θ phase. Simulated LFP obtained from the sum of RS membrane voltages. See Methods for details.

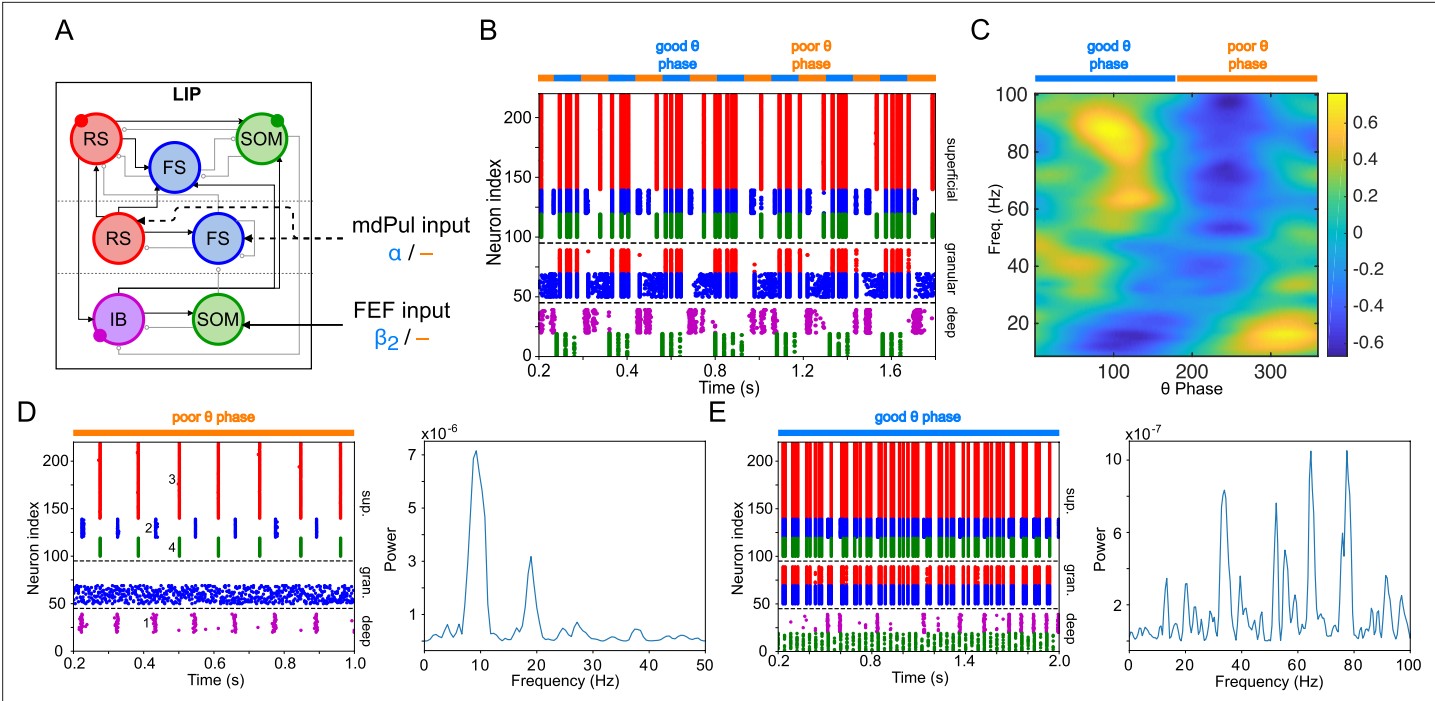

**Figure 3.** LIP module activity during the cue-target interval. (**A**). Diagram of the LIP module. Symbols are as in *Figure 1C*. (**B**). LIP module spiking during a 2 s long simulation with synthetic pulvinar and FEF inputs. (**C**). Spectral power as a function of $\theta$ phase for the simulation shown in (**B**). Simulated LFP obtained from the sum of RS membrane voltages. See Methods for details. (**D**). Reproduction of LIP $\beta_1$ rhythm characteristic of the poor $\theta$ phase. Left, raster plot of a 1 second simulation of LIP activity with no input, corresponding to the poor $\theta$ phase, showing a $\beta_1$ oscillatory rhythm obtained through period concatenation. A single cycle of this rhythm is labeled: 1, IB cells spike by rebound; 2, superficial FS cells spike in response to IB excitation; 3, superficial RS cells spike by rebound; 4, superficial SOM cells spike in response to RS excitation. Right, power spectrum of the LFP generated by superficial RS cells for the same simulation, with a peak in the $\beta_1$ frequency band. (Note that this simulation is half as long as those shown in **B** and **E**, to clearly show the structure of the $\beta_1$ oscillation.) (**E**). Reproduction of LIP γ rhythm, characteristic of the good $\theta$ phase. Left, raster plot of a 2 s simulation of LIP activity in the presence of inputs, corresponding to the good $\theta$ phase, showing γ rhythms in the granular and superficial layers. Right, power spectrum of the LFP generated by superficial RS cells for the same simulation, with peaks in the γ frequency band.

## Mechanisms of FEF and LIP oscillations during the cue-target interval

To model the period between cue and target presentation in the Egly-Driver task (*Fiebelkorn et al., 2013*), we introduced simulated mdPul input—consisting of bursts of α frequency (12 Hz) excitation arriving during half of each period of a (4 Hz) θ rhythm—targeting LIP granular layer RS and FS cells,

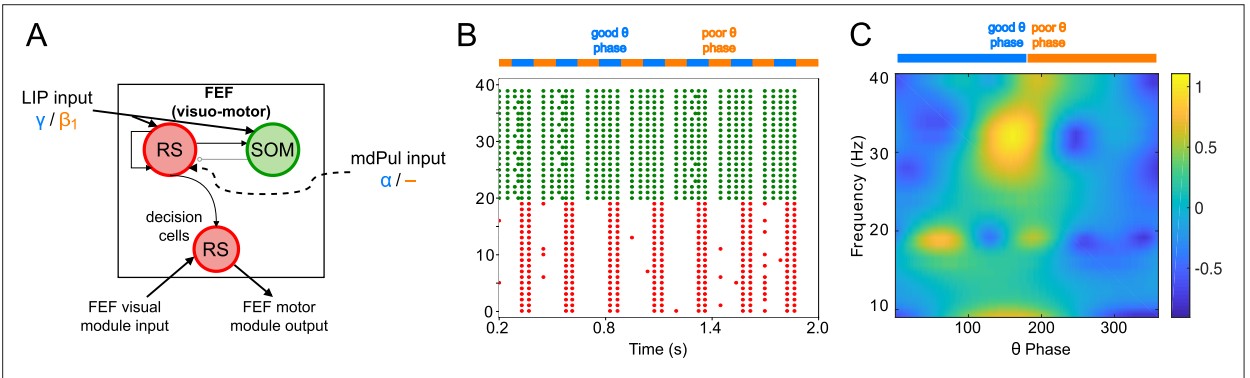

**Figure 4.** FEF visuomotor cell activity during the cue-target interval. (**A**). Diagram of the FEF visuomotor module. Symbols are as in *Figure 1C*. (**B**). Raster plot of a 1s-long simulation of FEF visuomotor cells with synthetic input. mdPul α input during the good $\theta$ phase alters the E-I balance, enabling RS and SOM cells to form a $\beta_2$ rhythm. No target was presented, so decision-cells were quiescent and are omitted. (**C**). Spectral power as a function of $\theta$ phase for simulation shown in **B**. Simulated LFP obtained from the sum of RS membrane voltages. See Methods for details.

and FEF visuomotor RS cells. Under these conditions, the model reproduced the nested rhythmic dynamics observed experimentally (*Figure 2B*): during mdPul excitation (the putative good θ phase), FEF visuomotor cells produced a $\beta_2$ rhythm, and LIP produced a γ rhythm; during mdPul quiescence (the putative poor θ phase), LIP produced a $\beta_1$ rhythm. Note that in the absence of additional visual input to FEF visual cells (i.e. when no target was present on screen), these cells and FEF decision cells (not shown in *Figure 2*) were relatively quiet. Below, we explain the circuit mechanisms of these rhythms and their interactions in detail for each module.

## LIP produces $\beta_1$ oscillations through period concatenation, and γ oscillations when stimulated by FEF and mdPul

The LIP module (*Figure 3A*) combined two previous laminar models of working memory and attention (*Kramer et al., 2008*; *Lee and Whittington, 2013*). The first of these was a model of a $\beta_1$ rhythm observed in rodent parietal slices (*Kramer et al., 2008*), shown to be useful for retention and manipulation of cell assemblies after the fading of a sensory input (*Gelastopoulos et al., 2019*). The model mimics experimental data (*Roopun et al., 2008*) in which adequate stimulation (sensory in vivo, excitatory agonist kainate in vitro) produced a γ rhythm in the superficial layers and an independent $\beta_2$ (~ 25 Hz) rhythm in the deep layers. Following a sufficient period of kainate application in vitro, the removal of kainate (modeling the disappearance of the sensory signal) caused both deep and superficial layers to switch to a $\beta_1$ (~ 15 Hz) rhythm (*Roopun et al., 2008*). A single period of this $\beta_1$ rhythm concatenated one γ period and one $\beta_2$ period (*Roopun et al., 2008*; *Kramer et al., 2008*), so that the duration of the $\beta_1$ period was the sum of the γ and $\beta_2$ periods.

Both the γ period and the $\beta_2$ period comprising the $\beta_1$ are initiated by rebound from inhibition; this allows the $\beta_1$ to remain active even at low levels of excitatory input (*Kramer et al., 2008*). As a consequence of the unusual physiology producing it, this rhythm exhibits unique functional properties: a superficial γ-rhythmic cell assembly excited by an initial stimulus maintains persistent activity as part of a $\beta_1$ rhythm after the cessation of the initial stimulus; and multiple cell assemblies can be added and subtracted from this 'working memory buffer' without interference (*Gelastopoulos et al., 2019*). Thus, this model is especially relevant to our task, which involves maintaining the cue location in working memory for up to $1.6s$, until target appearance.

Of note, the prior excitation required to produce the $\beta_1$ rhythm experimentally did not change the $\beta_2$ or γ rhythms during the period of excitation, but was hypothesized to produce plasticity altering interlaminar interactions. Such plasticity would occur in vivo either more rapidly during stimulation or in response to repeated stimulation. In computational models (*Kramer et al., 2008*; *Gelastopoulos et al., 2019*), the parameters were set as if this plasticity had already occurred; thus the shift from independent γ and $\beta_2$ rhythms to a $\beta_1$ rhythm required only a change model inputs.

The second inspiration for the LIP module was a model of the effects of (attentional) cholinergic modulation on the potentiation of γ power in the superficial layers of sensory cortex (*Lee and Whittington, 2013*). In this model, the excitation-inhibition balance in the granular layers is such that FS cells dominate RS cells, largely damping RS cell responses to excitatory input. In the presence of cholinergic modulation, β rhythms delivered to deep layers activate SOM interneurons, which in turn produce slowly decaying inhibition that suppresses granular layer FS cells. This results in disinhibition of granular layer RS cells, changing the excitation-inhibition balance sufficiently that RS cells drive FS cells to produce a γ rhythm. This input layer γ rhythm forms from the interaction of RS and FS cells, dependent on the FS cells receiving slow inhibition from deep layer SOM interneurons. In turn, granular γ rhythms drive γ rhythmicity in the superficial layers.

As in the working memory model, the circuitry of our LIP module produced a $\beta_1$ 'concatenation' rhythm in the absence of input from mdPul or FEF (i.e. in the poor θ phase, *Figure 3B–D*). We interpreted this $\beta_1$ as a 'post-excitation' rhythm, since in our model it always occurred following excitation—both that of cue presentation and that of the (prior) good θ phase. As in previous models, the excitation and the resulting plasticity were absorbed into the parameters of the model, which are set to the 'post-excitation' state throughout the simulation.

As in the previous model of attention (*Lee and Whittington, 2013*), $\beta_2$ input to deep SOM interneurons in our LIP module partially suppressed granular FS cells, leading to disinhibition of granular RS cells. (Granular FS cells remained active, but came under the control of granular RS cells.) When this disinhibition was combined with increased excitatory input to the granular layer, a γ rhythm resulted,

which then propagated to the superficial layer. During the good θ phase, the LIP module received exactly these two inputs—a $\beta_2$ input from FEF exciting deep SOM interneurons, and an α input from mdPul exciting the granular layer (*Figure 3A*). Thus, as in the previous model (*Lee and Whittington, 2013*), FEF beta input to the deep layers of LIP *created* the potentiated γ in the superficial layers of LIP during the good θ phase (*Figure 3B and C, E*; also see Robustness and Flexibility of the LIP Module).

The α-rhythmic input that excites the RS and FS cells of the LIP module's granular layer differed from the tonic input used previously (*Lee and Whittington, 2013*), but the result was the same. We modeled mdPul input excitation as a series of AMPA-like excitatory pulses arriving at α frequency. Since mdPul neurons are unlikely to fire in complete synchrony, the rise and decay time of the input synapses to LIP was increased (from $t_{rise} = 0.1ms$ and $t_{decay} = 0.5ms$ in AMPA synapses within the network, to $t_{rise} = 2ms$ and $t_{decay} = 10ms$ in input synapses, see *Appendix 2—figure 2A* ), although we found the superficial γ (but not its frequency) to be robust to changes in input synapse rise and decay times (*Appendix 2—figure 2*; see Robustness and Flexibility of the LIP Module for more details on the parameter-dependence of the superficial γ rhythm).

## FEF visuomotor circuits produce $\beta_2$ oscillations when stimulated by mdPul

The FEF visuomotor module consisted of RS cells reciprocally inhibited by SOM cells (*Figure 4A*). (FEF visuomotor module decision cells were active only when a target was present (see Mechanisms of Target Detection), and thus did not play a role in model behavior during the cue-target delay.) RS and SOM cells received excitatory inputs from LIP, and RS cells received excitatory inputs from mdPul. In line with experimental data (*Fiebelkorn et al., 2019c*) (and the behavior of our LIP module), synthetic LIP inputs to the FEF visuomotor module were γ frequency during the good θ phase and $\beta_1$ frequency during the poor θ phase. As in the LIP module, mdPul inputs at α frequency arrived only during the good θ phase. Excitatory noise, representing asynchronous inputs, also arrived from mdPul during the poor θ phase.

During the poor θ phase, the excitation-inhibition (E-I) balance between SOM and RS cells was such that the SOM cells (activated by tonic excitation) silenced the RS cells. During the good θ phase, RS cells, stimulated by mdPul, fired and took control of SOM cell activity. The timescale of SOM inhibition determined the frequency of the resulting RS-SOM rhythm, a $\beta_2$. This module exhibited a motif similar to that which created a γ in LIP: when SOM interneurons were too active, rhythms did not appear; when RS cells were stimulated, an altered E-I balance allowed rhythms (driven by RS cell firing) to emerge. Slow-decaying pulvinar stimulation (representing an input relying on a combination of NMDA and AMPA receptors) enabled up to two $\beta_2$ cycles to take place after each mdPul excitatory impulse. In the poor θ phase, the asynchronous input from pulvinar elicited a few spikes from RS cells, but this RS cell activity was not strong enough to initiate $\beta_2$ oscillations. *Figure 4B* shows a raster plot

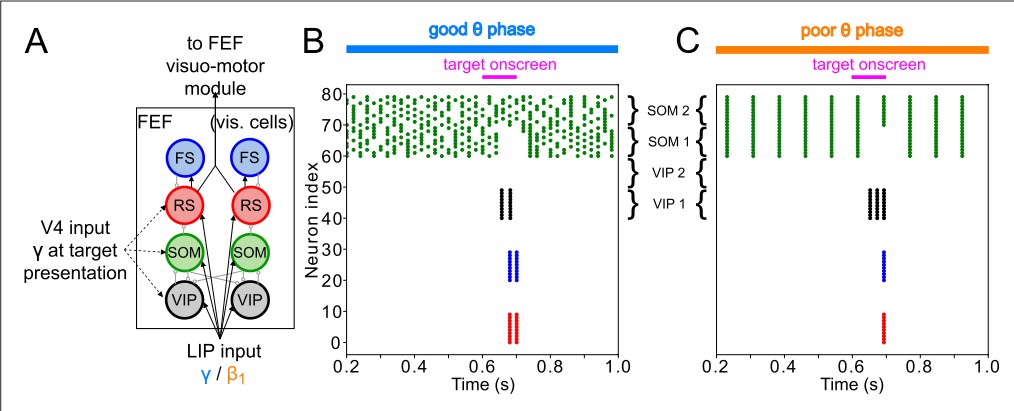

**Figure 5.** FEF visual module activity at target presentation. (**A**) Diagram of the FEF visual module. Symbols are as in *Figure 1C*. (**B**). Raster plot of a 1s-long simulation of FEF visual cells with good θ phase LIP input. A target appears on screen at $t = 600ms$ for a duration of $100ms$, producing two cycles of γ in RS and FS cells. (**C**). Raster plot of a 2 s simulation of FEF visual cells with poor θ phase LIP input. A target appears on screen at $t = 600ms$ for a duration of $100ms$, producing a single volley of RS spikes.

from a two second-long simulation of the isolated FEF visuomotor cell module, and a plot of its spectral power as a function of θ phase. Section Robustness and Flexibility of the FEF Visuomotor Module explores the dependence of FEF visuomotor module dynamics on model inputs and parameters.

## Mechanisms of target detection

### FEF visual VIP neurons detect small changes in visual input

The FEF visual module—consisting of RS, FS, VIP, and SOM cells (*Figures 1C and 5A*)—determined the full model's response to target presentation, by detecting the presence of a simulated target. We placed this target detection network in FEF visual cells to match current knowledge of the visual attention network. The appearance of a target on the screen was simulated as a transient ($100ms$) excitatory input to RS, VIP, and SOM cells at a γ frequency (50 Hz), presumably from an early visual area (e.g. V4, although another source, or excitation routed through LIP, are also possible).

The heart of the target detection circuit is the VIP–SOM interaction: both of these cells receive enhanced visual input upon target presentation. VIP and SOM cells are known to mutually inhibit each other (*Tremblay et al., 2016*), but while VIP have only local connections to SOM cells, SOM cells have global connections to VIP cells (*Karnani et al., 2016*; *Zhang et al., 2014*). This was instantiated in the module by separating the cells into two clusters, each representing a different location in visual space (*Figure 5A*). While SOM cells inhibited VIP cells from both clusters, VIP cells inhibited only SOM cells in the same cluster. RS and PV cells likewise connected only to other cells within the same cluster. The physiology of VIP interneurons—which exhibit large responses to relatively small inputs (*Lee et al., 2010*; *Wall et al., 2016*)—also played an important role in the model dynamics.

When no target was present on screen, all SOM, RS and VIP cells in the module received input from LIP. The sizes of the inputs were such that SOM cells were active, and SOM inhibition silenced VIP and RS cells. When a target appeared, a smaller γ input (coming from early visual areas, possibly via LIP) further excited the SOM, RS, and VIP cells of one cluster. Within this cluster, SOM cells, which were already highly active and exhibited a shallower f-I curve than the VIP cells (*Appendix 1—figure 1*; *Wall et al., 2016*; *Millman et al., 2020*), did not change their spiking pattern. The VIP cells in this cluster therefore received the same amount of inhibition, but a higher amount of excitation, which, due to their steeper f-I curve (*Appendix 1—figure 1*; *Wall et al., 2016*; *Millman et al., 2020*), caused them to fire. VIP firing then temporarily silenced SOM cells, disinhibiting RS cells and enabling them to fire. Though such increased excitability of RS cells could be obtained though many different mechanisms, the interaction of VIP and SOM cells amplified a small difference in excitation between two visual regions, resulting in a large RS cell response to even a small increase in excitation. We hypothesize that these mechanisms play an important role in detecting near-threshold visual stimuli, such as the low contrast target of the task we are modeling. We discuss the robustness of this module in more detail in Modeling Choices in the FEF Visual Module, and its relation to other models of VIP–SOM interactions in Discussion: Modeling Choices & Limitations.

In this module, the influence of θ rhythm phase was via LIP input, which was a γ rhythm in the good θ phase and a $\beta_1$ rhythm in the poor θ phase. During the good θ phase, high-frequency input and the interaction of RS and FS cells produced a burst of γ rhythmic activity in response to target appearance (*Figure 5B*). During the poor θ phase, single volleys of LIP input produced single volleys of FEF visual RS cell spikes (*Figure 5C*). Note that target detection in the FEF visual cell module is not yet adequate for a 'hit' to be recorded for the model. This module's response to a target appearing on the screen still needs to be transmitted to the FEF visuomotor module, where a decision to act on it is made by what we call 'decision cells'. Only the spiking of decision cells following target appearance was considered a 'hit'.

### FEF decision cells enable action following target detection

The output of FEF visual and visuomotor RS cells was transmitted to a set of RS cells in the visuomotor module that we refer to as decision cells, whose spiking activity we interpreted as initiating action in response to target detection. Input from (non-decision-related) FEF visuomotor RS cells set the threshold for decision cell spiking, determining how likely they were to respond to (simultaneous) input from FEF visual cells during target presentation. There is evidence to suggest that FEF motor neurons participate in overt and covert visual responses to the target (*Bruce and Goldberg, 1985a*; *Buschman and Miller, 2009*); these cells can receive inputs from other brain areas, allowing them to

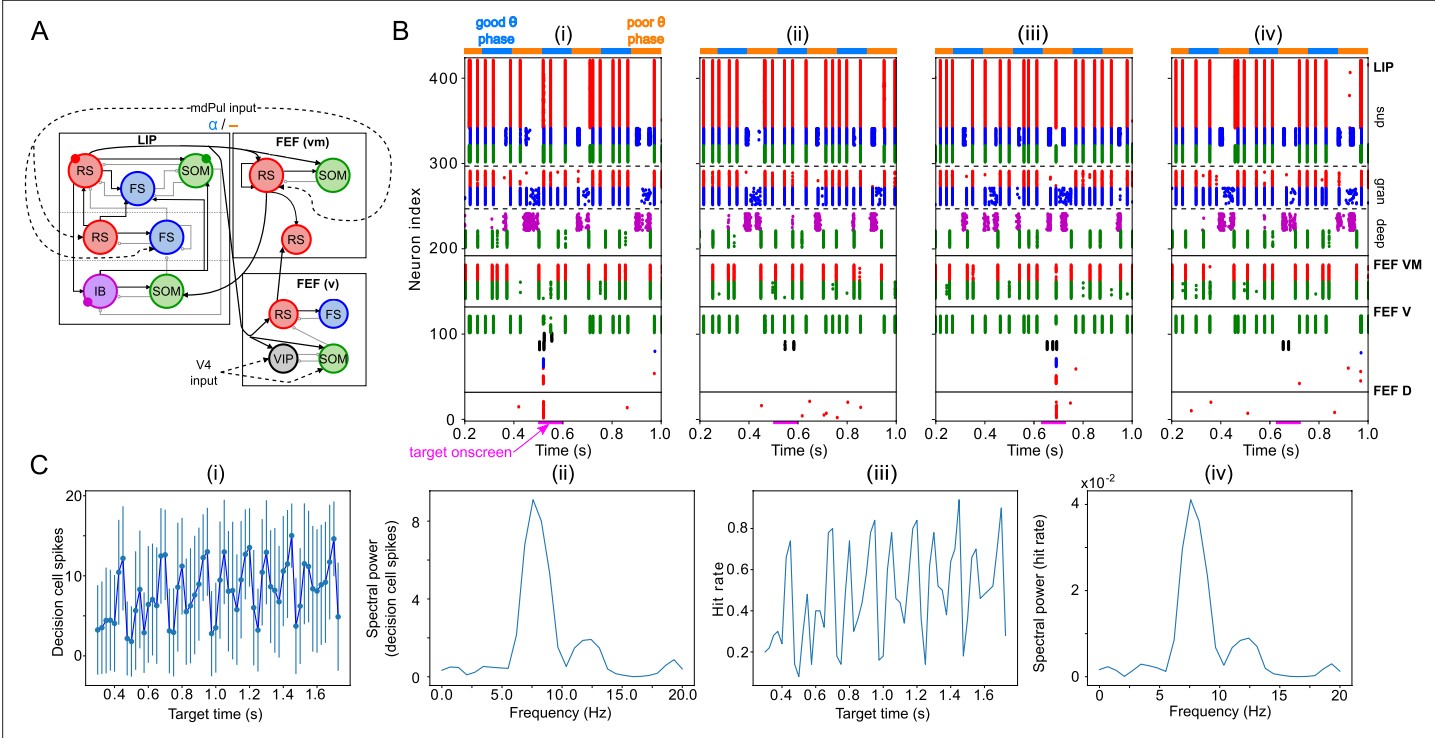

**Figure 6.** Model activity in response to target presentation. (**A**). Diagram of the model. Symbols are as in *Figure 1C*. (**B**). Raster plots of the model during a 1 s long simulation with simulated mdPul input and target presentation during the good or poor $\theta$ phase. The raster plots show activity of LIP on top, that of FEF visuomotor cells right below, that of FEF visual cells below them, and that of FEF decision cells at the bottom. A target was presented for 100ms in the good $\theta$ phase in (**i**) and (**ii**), and in the poor $\theta$ phase in (**iii**) and (**iv**). Only in (**i**) and (**iii**) did target presentation lead to a 'hit'. (**C**). Decision cell activity and hit rate depend on $\theta$ phase. (**i**) Number of decision cells spikes (with mean and standard deviation) during target appearance as a function of the cue-target interval. (**ii**) Power spectrum of decision cell spiking, showing a peak at a $\theta$ frequency (8 Hz). (**iii**) Hit rate as a function of the cue-target interval. (**iv**) Power spectrum of the hit rates, showing a peak at a $\theta$ frequency (8 Hz).

initiate task-appropriate responses (e.g. covert attentional shifts or eye movements). Thus, we assume decision cell spiking would be relayed to FEF motor cells to trigger action initiation (not modeled here).

## Task performance

### Network response to target appearance is greater during the good $\theta$ phase

To test whether the network dynamics observed during good and poor θ phase also affect target detection, we simulated the activity of the entire network for 2 s, presenting a simulated target (modeled as a 100ms-long train of γ frequency excitatory pulses to FEF visual neurons) at varied times distributed across several θ cycles. (Note that the phase relationships between different rhythms and modules are now determined by intrinsic dynamics rather than predetermined by the mdPul θ rhythm.) The model's response to the target depended on its timing relative to the ongoing θ cycle (*Figure 6*). More precisely, the number of FEF decision cell spikes following target appearance was higher on average when the target appeared (i.e. when target input began) during the good θ phase (or at the transition from poor to good θ phase) compared to when it appeared during the poor θ phase (or at the transition from good to poor θ phase; *Figure 6C* (i)).

We translated FEF decision cell spiking into a hit rate via thresholding: we considered a trial to be a 'hit' if more than half of FEF decision cells spiked at least once during the $100ms$-long window where the target was presented. On individual trials, hits and misses occurred during both good and poor θ phases (*Figure 6B*). In agreement with behavioral data, model hit rates varied with the cue-target interval; somewhat surprisingly, they also agreed with behavioral results in exhibiting an 8 Hz θ rhythm (*Figure 6C* (iii) & (iv)).

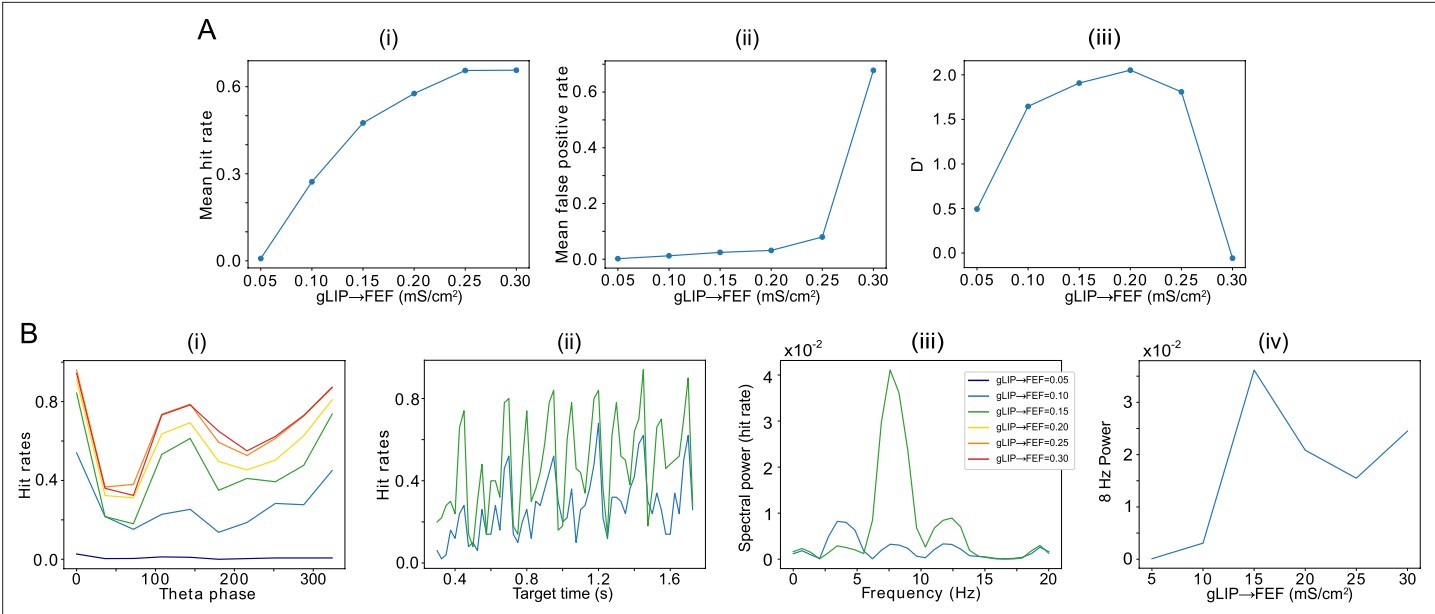

**Figure 7.** Effect of the strength of LIP -> FEF chemical synapses on network sensitivity and hit rates rhythmicity. (**A**).The strength of LIP → FEF chemical synapses influences network sensitivity. (**i**) Hit rate as a function of $g_{LIP \to FEFv}$. (**ii**) False alarm rate as a function of $g_{LIP \to FEFv}$. (**iii**) D' as a function of $g_{LIP \to FEF}$. (**B**). Increased $g_{LIP \to FEFv}$ leads to a stronger 8 Hz component in the hit rates. (**i**) Hit rate as a function of $\theta$ phase plotted for values of $g_{LIP \to FEFv}$ between 0.05 and 0.3. (**ii**) Hit rate as a function of the cue-target interval for $g_{LIP \to FEFv} = 0.10 mS/cm^3$ and $g_{LIP \to FEFv} = 0.15 mS/cm^3$. (**iii**) Power spectra of the hit rate time series shown in (**ii**). (**iv**) 8 Hz power as a function of $g_{LIP \to FEFv}$.

## LIP→FEF functional connectivity influences network sensitivity and hit rate rhythmicity

We varied the functional connectivity from LIP to the FEF visual module by altering the strength of the chemical synapses from superficial LIP RS cells to FEF visual cells. Increasing LIP→FEF connectivity increased both hit rates (*Figure 7A* (i)) and false alarm rates (*Figure 7A* (ii)). (We considered a trial to contain a 'false alarm' whenever more than half of the decision cells fired in any $25ms$-long window outside target presentation.) Computing the D' sensitivity index, which combines both hit and false alarm rates, revealed that intermediate values of LIP→FEF connectivity resulted in the highest levels of task performance (i.e. both high hit rates and low false alarm rates, *Figure 7A* (iii)).

The LIP→FEF connectivity was also related to the 8 Hz rhythmicity observed in the dependence of the hit rate on the cue-target interval. For sufficiently high values of the conductance of LIP→FEF chemical synapses, the temporal profile of hit rates exhibited a dominant 8 Hz component (*Figure 7B*), while for small values of the LIP→FEF conductance, the hit rates exhibited robust 4, 8, and 12 Hz periodicity. Varying LIP→FEF conductance revealed that, while a hit rate peak at the beginning of the good θ phase (at 0 degrees) was common across 4 and 8 Hz hit rate rhythms, 8 Hz rhythms resulted from the addition of another, smaller peak in hit rate at the beginning of the poor θ phase (*Figure 7B* (i) & (ii)). This was a result of LIP superficial RS spikes participating in $\beta_1$ oscillations, which excited FEF visual neurons, and rescued target detection in the poor θ phase. Plotting the number of decision cell spikes during target presentation as a function of the phase of the LIP $\beta_1$ oscillation in the poor θ phase revealed a stronger influence of LIP $\beta_1$ phase on decision cell spiking for a medium than for a low value of the LIP→FEF synaptic conductance (*Appendix 1—figure 2A*). The LIP→FEF synaptic conductance also influenced the dependence of the hit rate on the LIP $\beta_1$ phase (*Appendix 1—figure 2B*).

The dominance of the 8 Hz rhythm in hit rate for high LIP→FEF connectivity was also dependent on the shape of mdPul input to LIP and FEF. When mdPul input had the same maximal conductance across the entire good θ phase, the power of the 8 Hz component to the hit rate rhythm was much lower, and it was comparable in size to the 4 Hz component in the hit rate rhythm for all values of LIP→FEF connectivity (*Appendix 1—figure 3A*). An increase in 8 Hz power with the strength of LIP→FEF connectivity was still observed, however (*Appendix 1—figure 3A*). Notwithstanding this

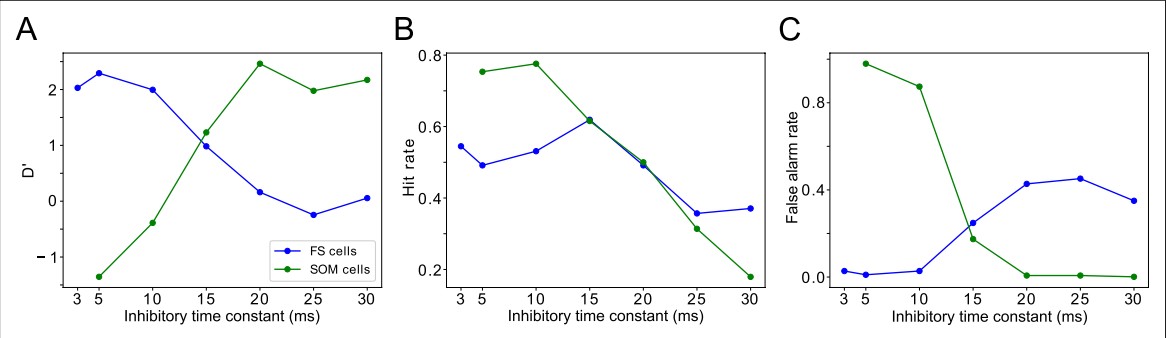

**Figure 8.** Changing timescales of FEF and LIP rhythmic dynamics negatively impacts task performance. (**A**). D' measure as a function of the time constants of inhibition for SOM cells and FS cells. (**B**). Hit rate as a function of the time constants of inhibition for SOM cells and FS cells. (**C**). False alarm rate as a function of the time constants of inhibition for SOM cells and FS cells.

difference in the dominance of 8 Hz rhythmicity, the dependence of network sensitivity on LIP→FEF connectivity (and in particular the value of $g_{LIP \to FEF}$ resulting in peak sensitivity) remained unchanged for constant mdPul input (*Appendix 1—figure 3B*).

In behavioral data (*Fiebelkorn et al., 2013*), an 8 Hz rhythm in hit rate was observed for trials in which the target and cue appeared within the same object, while a 4 Hz rhythm in hit rate was observed for trials in which the target and cue appeared within different objects. Our model is consistent with the hypothesis that the deployment of attention towards a cued object moderately increases the strength of LIP→FEF connectivity in neural populations representing that object, resulting in both an increase in performance (as reflected by an increase in D') and a change in the rhythmicity of behavior.

## Sensitivity of target detection is highest for FEF oscillations at β frequency and LIP oscillations at γ frequency

One question that naturally arose from the previous results was whether the frequency of LIP and FEF rhythms is significant for task performance. To answer it, we modified the synaptic time constants of inhibition in FS cells and SOM cells ($\tau_{d,FS}$ and $\tau_{d,SOM}$, respectively), varying the frequencies of inhibition-paced rhythms across the β and γ ranges. We then analyzed the temporal profile of hit rates as a function of the cue-target interval, as in the previous section.

We found the sensitivity of target detection (as measured by D') was highest for $\tau_{d,SOM} \simeq 20$ ms, corresponding to observed $\beta_2$ frequencies of ~30 Hz, and for $\tau_{d,FS} \simeq 5$ ms, corresponding to observed γ frequencies of 40 Hz and above (*Figure 8A*). The peak in sensitivity at these time constants resulted from shifts in both hit rates and false alarm rates (*Figure 8B & C*). We describe the mechanisms of these changes in detail below. A key point is that the activation of FEF visual module SOM cells at a γ timescale, that is, faster than their β time constant of inhibition, provides a blanket of inhibition that prevents VIP cells from spiking outside of target presentation and reduces false alarms in the network during the good θ phase.

### Effects of decreased inhibitory time constant of SOM cells

Both the hit and false alarm rates trended downwards as $\tau_{d,SOM}$ increased, but the false alarm rate decreased more rapidly initially and then bottomed out at nearly zero for $\tau_{d,SOM}$ around 20ms (*Figure 8B & C*). As a result, $\tau_{d,SOM} < 20ms$ resulted in proportionally more false alarms, while $\tau_{d,SOM} > 20ms$ resulted in proportionally fewer hits, as compared to the highest observed sensitivity.

Because SOM cells did not exhibit self-inhibition, the rate of SOM cell firing (and the time between volleys in SOM population spiking) was controlled by the strength and rate of their input. Whether $\tau_{d,SOM}$ was longer or shorter than the period between SOM cell volleys controlled the proportion of the time that SOM cell targets were inhibited. Thus, when $\tau_{d,SOM}$ increased from 5 to 30ms, the overall firing rates of RS cells in all network modules decreased (*Figure 9A*). For the same reason, the activation of FEF visual module SOM cells at a timescale faster than $\tau_{d,SOM}$ was key to preventing false alarms in the network during the good θ phase, as it provided a sufficient blanket of inhibition to prevent VIP cells from spiking outside of target presentation. As $\tau_{d,SOM}$ increased, this blanket of

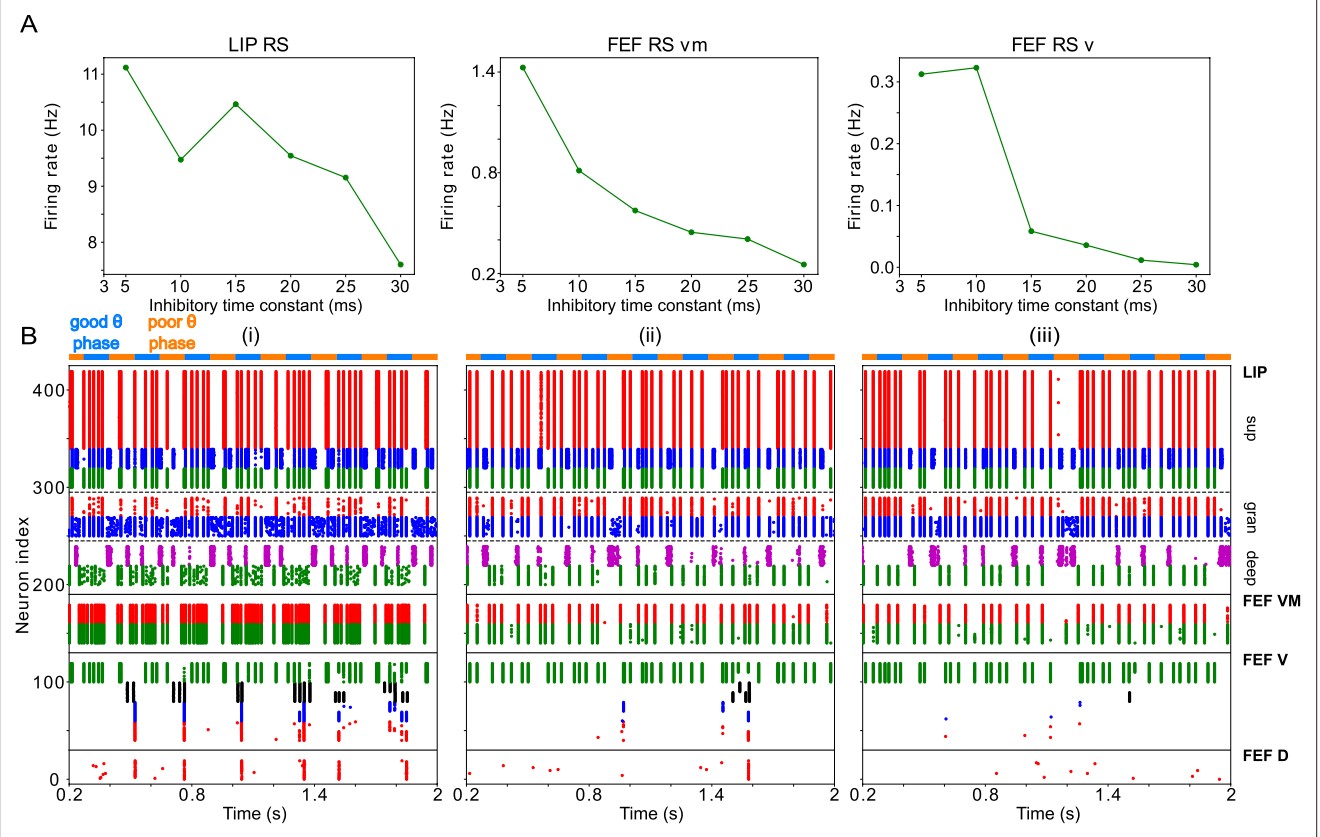

**Figure 9.** Changing $\tau_{d,SOM}$ impacts network dynamics and function. (**A**) The firing rates of RS cells in the LIP, FEF visuomotor, and FEF visual modules as a function of $\tau_{d,SOM}$. (**B**) Raster plots of network activity for three values of $\tau_{d,SOM}$: (**i**) $\tau_{d,SOM} = 5$ ms, showing several false alarms; (**ii**) $\tau_{d,SOM} = 20$ ms, showing a hit; (**iii**) $\tau_{d,SOM} = 25$ ms, showing a miss.

inhibition became stronger, and thus a stronger response from the FEF visual cells was needed to register a hit.

Thus, $\tau_{d,SOM}$ affected the decision criterion of the network—how likely the network was to register a hit in any context—with longer $\tau_{d,SOM}$ resulting in a higher decision criterion. Because target presentation was accompanied by extra input to FEF visual cells, decreasing $\tau_{d,SOM}$ from its longest value of 30ms resulted in an increase in the hit rate even though the blanket of inhibition was still intact (*Figure 9B*). Once $\tau_{d,SOM}$ fell below 20ms, the blanket of inhibition developed 'holes', and the false alarm rate began increasing (*Figure 9B*). Because a false alarm could occur at any time, while a hit could only occur when a target was present, the rate of false alarms eventually increased much more quickly than the hit rate.

## Effects of increased inhibitory time constant of FS cells

The hit and false alarm rates both changed non-monotonically with $\tau_{d,FS}$, and network activity depended in a more complex way on this parameter (*Figures 8B & C , and 10*). Nevertheless, the changes in hit rate were relatively small as $\tau_{d,FS}$ was varied, and thus the sensitivity of the network was dominated by the changes in false alarm rate as $\tau_{d,FS}$ increased, with the lowest false alarm rates and the highest D' occurring for $\tau_{d,FS} = 5$ ms.

FS cells provided inhibition to RS cells in the FEF visual module and in LIP granular and superficial layers. Unlike SOM cells, FS cells exhibited self-inhibition. Thus, increasing $\tau_{d,FS}$ produced longer lasting inhibition in FS cells as well as their inhibitory targets. In LIP granular and superficial layers, where the dominant inputs to RS and FS cells are the same, RS cell spiking decreased nearly monotonically with $\tau_{d,FS}$ (*Figure 10A*). In the FEF visual module, the situation was more complicated. The rate of FEF visual RS cell firing was controlled by $\tau_{d,FS}$ both directly, through the slowing of recurrent FS inhibition, and indirectly, through a change in LIP input to FEF visual VIP, SOM, and RS cells. As

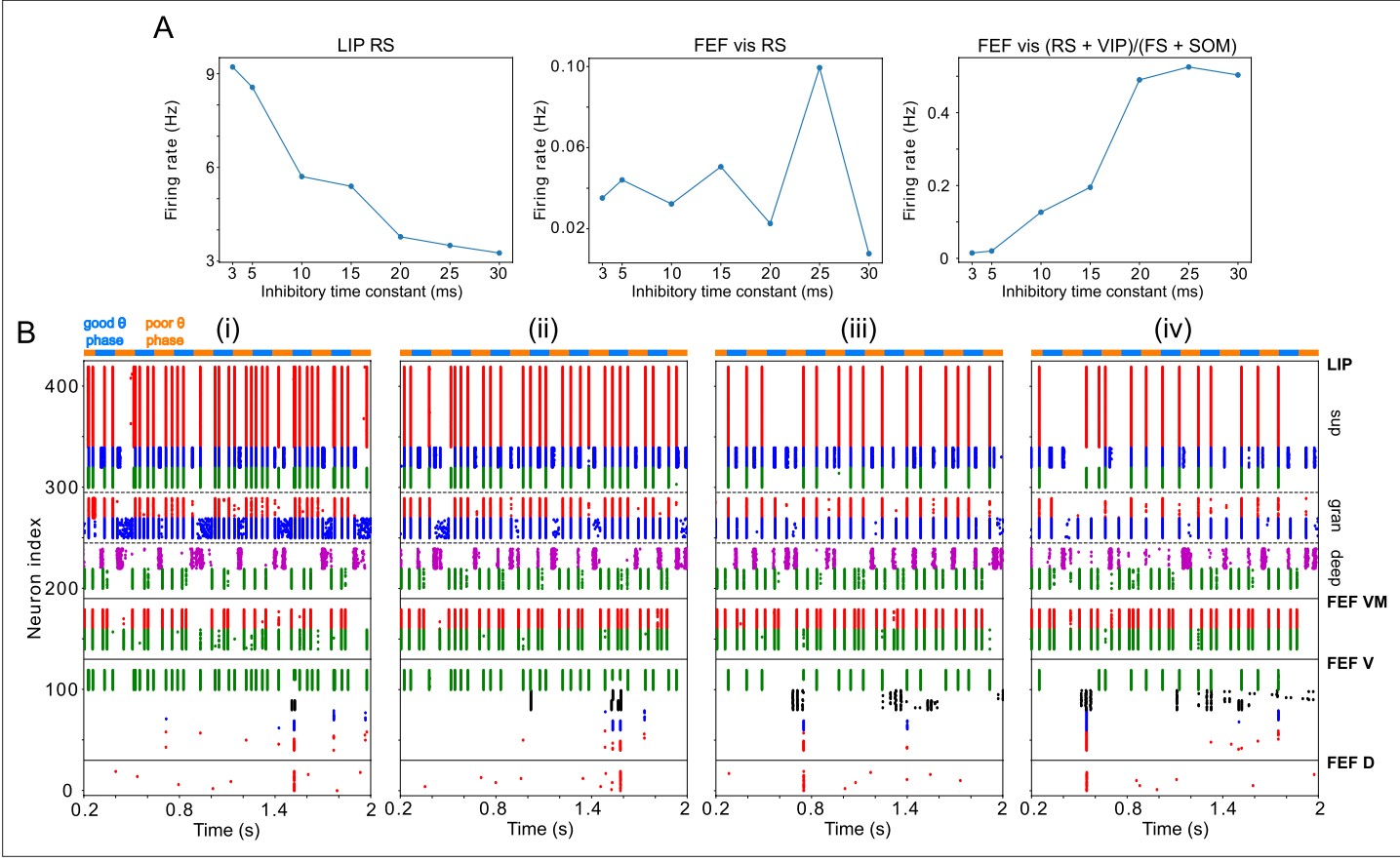

**Figure 10.** Changing $\tau_{d,FS}$ impacts network dynamics and function. (**A**) The firing rates of RS cells in the LIP superficial layer and the FEF visual module, and the sum of excitatory and disinhibitory spike rates divided by inhibitory spike rates, as a function of $\tau_{d,FS}$. (**B**) Raster plots of network activity for four values of $\tau_{d,FS}$: (**i**) $\tau_{d,FS} = 5$ ms (showing a hit); (**ii**) $\tau_{d,FS} = 10$ ms (showing a hit); (**iii**) $\tau_{d,SOM} = 20$ ms (showing a false alarm); (**iv**) $\tau_{d,FS} = 30$ ms (showing a false alarm).

a result, changes in the hit and false alarm rates did not mirror changes in FEF visual RS cell activity (*Figure 10A*). A plot of the sum of RS and VIP activity divided by the sum of FS and SOM cell activity came closer to matching the changes in false alarm rate and sensitivity, by the logic outlined above (i.e., increases in excitation/disinhibition relative to inhibition leading to increases in the false alarm rate; *Figure 10A*). However, a full understanding of why the network sensitivity is highest for $\tau_{d,FS} = 5$ ms will require further work.

## Model robustness and flexibility

As shown in the previous sections, our model reproduced both the neural dynamics and the behavioral results observed during rhythmic sampling in spatial attention (*Fiebelkorn et al., 2019c*). In order to be relevant to the brain, our model must be somewhat robust to parameter changes. However, since LIP and FEF are involved in cognitive tasks other than rhythmic attention, it should also be flexible enough to allow different rhythms to appear as conditions are varied. This section will address both these issues by looking at the conditions necessary for oscillations and target detection in the different submodules of the model.

### Robustness and flexibility of the LIP module

The activity of the LIP module depended on both inputs from FEF and mdPul and tonic excitation to its granular layer (see *Appendix 2—figure 1* and Influence of Tonic Excitation on LIP Behavior). While α input from mdPul helped sustain γ oscillations, it did not play as important a role as the disinhibition mediated by the (top-down) $\beta_2$ from FEF (*Figure 11*). When FEF $\beta_2$ input to LIP deep layers was absent, deep SOM interneurons remained quiescent, granular layer FS cells remained tonically

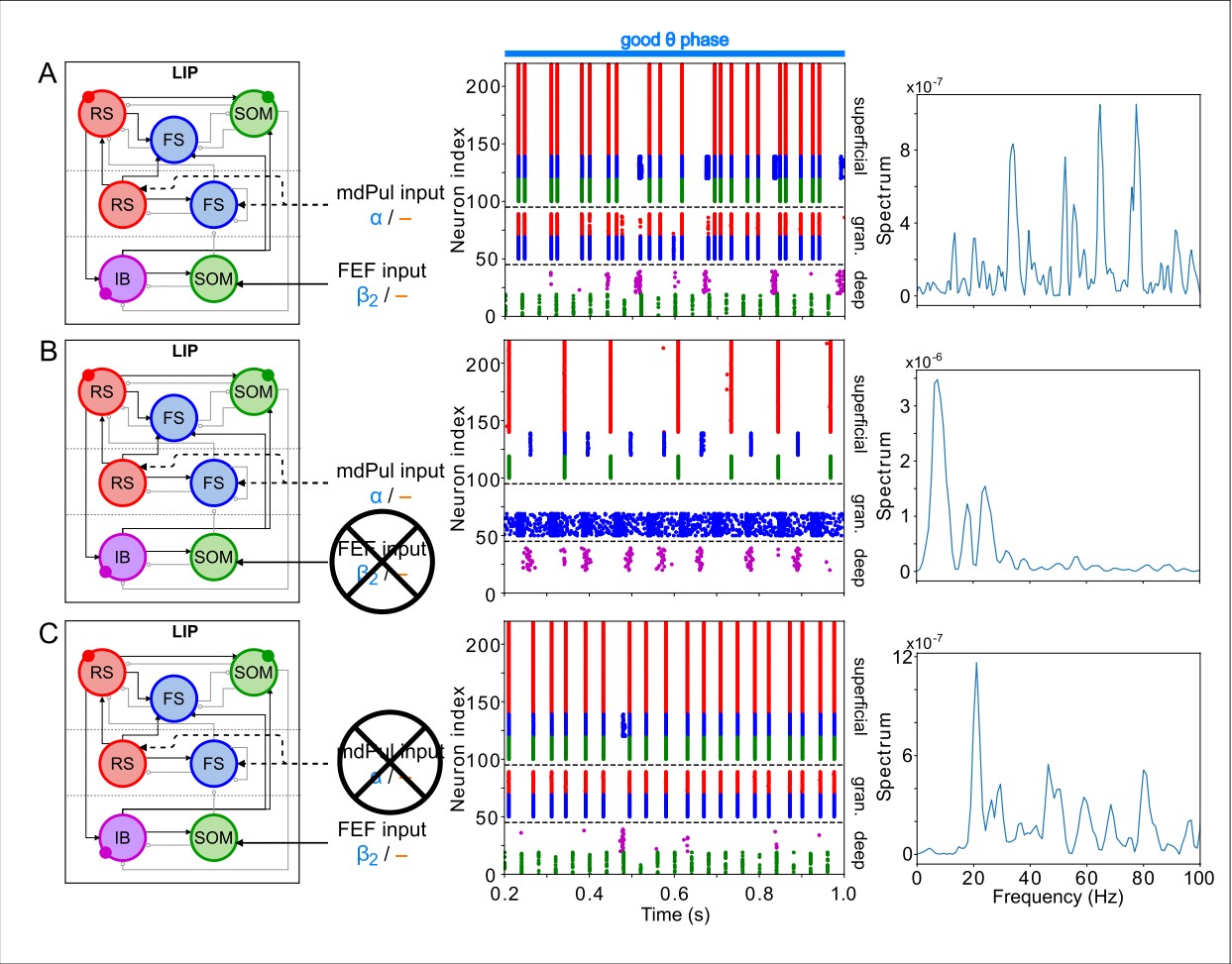

**Figure 11.** LIP good $\theta$ phase behavior depends on mdPul and FEF inputs. (**A**). LIP good $\theta$ phase behavior with both mdPul and FEF inputs. (**B**). LIP good $\theta$ phase behavior with FEF input but no mdPul input. (**C**). LIP good $\theta$ phase behavior with mdPul input but no FEF input. Columns are as follows: Left, diagram of the model. Symbols are as in *Figure 1C*. Middle, raster plot of a 1s-long simulation. Right, power spectrum of the LFP generated by the network in this 1s-long simulation.

active, and granular layer RS cells receiving tonic FS inhibition remained quiescent. The superficial and deep layers then interacted to produce a $\beta_1$ rhythm, just as in the poor θ phase (*Figure 11B*). On the other hand, when mdPul α input was absent, granular RS cells continued to spike when disinhibited by the $\beta_2$ drive to LIP deep layer SOM cells, although at a lower β frequency ($\sim$ 20 Hz, *Figure 11C*). Additionally, modeling mdPul synapses with a shorter rise and decay time did not prevent the LIP module from generating a superficial γ rhythm (*Appendix 2—figure 2B*), though they reduced its frequency; this γ rhythm also persisted when the rise and decay times of mdPul synapses were increased (*Appendix 2—figure 2C*).

LIP rhythms did depend on the level of tonic excitation to the granular layer (*Appendix 2—figure 1*). It is noteworthy that though the behavior of the LIP module was robust to some changes to the tonic excitation level, sufficient excitation produced qualitatively different dynamics in LIP with consequences for the behavior of the entire network, which we hypothesized might be functional for other cognitive tasks. For example, LIP's ability to direct attention to a highly salient stimulus in a bottom-up fashion (*Bisley et al., 2011*; *Buschman and Miller, 2007*; *Miller and Buschman, 2013*) was reproducible in our model by supplying sufficiently strong stimulation to the LIP granular layer (presumably coming from V4), which initiated a γ rhythm even in the absence of mdPul and FEF input (*Appendix 2—figure 1*; see Robustness and Flexibility of the FEF Visuomotor Module and Discussion for more details).

## Robustness and flexibility of the FEF visuomotor module

Similarly to LIP, the activity of the FEF visuomotor module depended on rhythmic input received from LIP and mdPul, as well as on tonic excitation to its RS and SOM cells. Pulvinar input was required for the network to achieve a $\beta_2$ rhythm during the good θ phase—without mdPul input, RS cells spiked only once during each good θ phase (*Appendix 3—figure 1A*). The frequency of mdPul input also played a role in generating sustained $\beta_2$ oscillations: input at θ or slower frequencies did not provide enough excitation to RS cells for them to take control of SOM cells during the entire good θ phase (*Appendix 3—figure 1B*). Model behavior was preserved when mdPul input arrived at frequencies faster than α, but not when the width of mdPul excitatory pulses was varied significantly (*Appendix 3—figure 1B, C*). The frequency of the alternation of the good and poor θ phases was required to be less than half of mdPul α frequency for the network to achieve a $\beta_2$ rhythm in the good θ phase (*Appendix 3—figure 1D*).

LIP input to RS and SOM cells also influenced the activity of the FEF visuomotor module. Simply removing this input decreased SOM inhibition of RS cells, causing them to fire at a rhythm faster than the $\beta_2$ experimentally observed during the good θ phase, and also during the poor θ phase (*Appendix 3—figure 2A*). For the same reason, LIP input at a frequency in the α range or lower led to FEF visuomotor cells exhibiting a faster rhythm in the good θ phase (*Appendix 3—figure 2B*). LIP input at a $\beta_2$ or γ frequency was required to achieve the desired $\beta_2$ frequency in FEF visuomotor RS cells. Increasing the frequency of LIP input above the γ range caused FEF visuomotor RS cells to fire at a frequency faster than $\beta_2$, provided that the input was strong enough. Because FEF visuomotor RS cell spiking could occur even if LIP was still producing the $\beta_1$ rhythm characteristic of its poor θ phase dynamics, our model suggests that the transition from the poor to the good θ phase is likely initiated by FEF, which then recruits LIP in a top-down fashion.

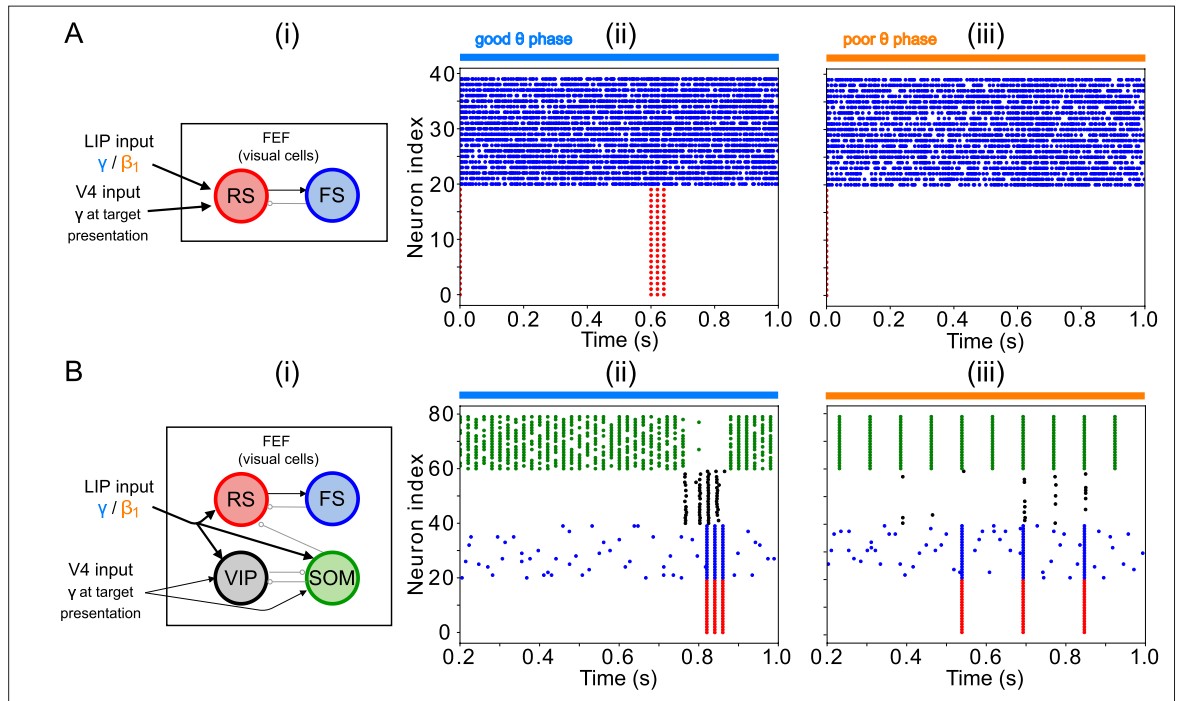

**Figure 12.** Modeling choices in the FEF visual module. (**A**). A model without VIP and SOM cells fails to detect a small change in input. (**i**) Diagram of FEF visual module without VIP and SOM cells. Symbols are as in *Figure 1C*. (**ii**) Raster plot of a 1 s simulation of the model with a target input applied between 600ms and 700ms at 1.5 times the strength of the background input, producing a γ burst in RS cells. (**iii**) Raster plot of a 1 s simulation of the model with a target input applied between 600ms and 700ms at 0.5 times the strength of the background input, producing no activity in RS cells. (**B**). A model with a single population leads to false alarms in the poor θ phase. Panel (**i**) as above. (**ii**) Raster plot of a 1 second simulation of the model with good θ phase inputs, and target input applied between 600ms and 700ms, leading to γ activity in RS cells only during target presentation. (**iii**) Raster plot of a 1 s simulation of the model with poor θ phase inputs, and target input applied between 600ms and 700ms, leading to RS cell activity outside of target presentation.

Even at experimentally observed frequencies, the strength of LIP input influenced FEF visuomotor rhythms (*Appendix 3—figure 2C*). A too-weak input did not provide enough excitation to FEF SOM cells to silence FEF during the poor θ phase, and resulted in FEF to producing a rhythm faster than $\beta_2$ during the good θ phase, while with a too-strong input FEF RS cell spiking followed the LIP input rhythm (at a $\beta_1$ frequency during the poor θ phase, and a γ frequency in the good θ phase). Importantly, although this behavior is undesirable for the rhythmic attention task we are modeling here, the activation of the FEF visuomotor module by LIP input in the absence of mdPul stimulation may represent a bottom-up attentional state, in which the network responds to the appearance of a highly salient target in a bottom-up fashion.

Finally, just as LIP activity depended on the tonic excitation given to its input layer RS and FS cells, FEF visuomotor activity depended on the tonic excitation given to its RS and SOM cells (*Appendix 3—figure 3*).

## Modeling choices in the FEF visual module

While simpler versions of the FEF visual module were able to detect changes in contrast, they exhibited some disadvantages that motivated our modeling choices for this module (*Figure 12*). A model including only RS and FS cells (*Figure 12A*) exhibited a γ burst during target presentation in response to strong-enough excitation of RS cells, even without the disinhibition provided by VIP cells. However, since RS cells have a larger capacitance than VIP cells, the strength of the target input had to be at least 1.5 times larger than the background stimulation from LIP to result in γ rhythmicity (*Figure 12A* (ii) & (iii)), which we interpreted as a high contrast target. Thus, the VIP and SOM cell circuit in the visual module played an important role in the detection of near-threshold contrast changes in our simulations.

In contrast, simulations of a model including only one interconnected network of RS, FS, VIP, and SOM cells (as opposed to two clusters of cells with different inter- and intra-cluster connectivity) were unable in our hands to satisfactorily exhibit robust target detection (*Figure 12B*). Robust target detection depended on VIP cells being silenced by SOM cells when no target was present, and SOM cells being silenced by VIP cells during target presentation, during both the good and the poor θ phase (that is, with background inputs from LIP in the γ or $\beta_1$ range). We were unable to achieve this behavior for any of the conductances of VIP→SOM inhibitory synapses or levels of tonic excitation to VIP and SOM cells tested. An example of an unwanted behavior in a model without clusters is shown in *Figure 12B* (ii) & (iii), where the inhibition from SOM to VIP was strong enough to ensure VIP cells didn't spike outside target presentation in the good θ phase (that is, when LIP input was in the γ range and SOM cells are also active at a γ rhythm), but not in the poor θ phase (when LIP input and SOM cell spiking were at $\beta_1$ frequency). Increasing the inhibition from VIP to SOM led to an inability of VIP cells to spike at target presentation. Separating SOM and VIP cells into two clusters with differential intra- and inter-cluster connectivity enabled a finer tuning of the inhibition between them (in particular, VIP cells received inhibition from only half of the SOM population, rather than all-or-nothing SOM inhibition), which helped achieve the dynamics necessary for task performance under a variety of different constraints.

## Discussion

We constructed a biologically detailed model of LIP and FEF circuits capable of reproducing the complex oscillatory dynamics observed at a cued location during a sustained attention task (*Fiebelkorn et al., 2019c*). The model was driven by rhythmic pulvinar input, the presence and absence of which defined periods of relatively enhanced and diminished sensory processing, the 'good' and 'poor' θ phases observed experimentally (*Fiebelkorn et al., 2013*; *Fiebelkorn et al., 2019c*). An LIP module with a laminar structure capable of producing $\beta_1$ and γ oscillations was motivated by previous work on memory and attention (*Kramer et al., 2008*; *Lee and Whittington, 2013*). FEF visuomotor and visual modules produced $\beta_2$ oscillations and detected the appearance of a target on a screen, respectively.

In this model, biophysically detailed rhythmic dynamics interacted to support enhanced sensitivity in visual target detection during the good θ phase, while maintaining functional flexibility—for example, the potential for target detection during the poor θ phase. The incorporation of mechanisms

originally developed to illuminate brain function in related but distinct cognitive contexts provided a further constraint on the realism and flexibility of the model.

Our work suggests that while pulvinar primes LIP and activates FEF, it is FEF that elicits γ-rhythmic output from LIP in the context of top-down attentional control via $\beta_2$ oscillatory input. It shows that the frequencies of observed rhythms maximize these interactions and the sensitivity of target detection, suggesting that these rhythms, rather than being epiphenomena, are key to the function of attention in the brain. It suggests that LIP→FEF functional connectivity (operationalized as the maximal synaptic conductance) may contribute to the experimentally observed shift in hit rate periodicity from 4 Hz (for the uncued object, and reflecting the frequency of mdPul input) to 8 Hz (for the cued object). Finally, our model suggests hypotheses about how multiscale rhythmic coordination may serve to sequence and segregate the activities of perceptual and motor systems. We elaborate on these points below.

## Interacting rhythms in FEF and LIP mediate the competition between top-down and bottom-up Control of visual attention

The rhythms in our network did not operate independently. The FEF visuomotor $\beta_2$ oscillation—activated by pulvinar excitation of FEF visuomotor cells in the good θ phase—elicited γ oscillations from LIP by effectively disinhibiting LIP granular layer pyramidal cells. LIP γ oscillations in turn acted to enhance sensory processing in the network by stimulating FEF visual cells, making them more likely to detect a target. This increased hit rates during the good θ phase, as did the direct excitation of decision cells by FEF visuomotor $\beta_2$ rhythms. In the context of our task (in which the LIP granular layers did not receive very strong inputs), the FEF $\beta_2$ rhythm was necessary for the production of LIP γ. In contrast, the FEF RS cell spiking characteristic of the good θ phase could occur even if LIP was still producing the $\beta_1$ characteristic of its poor θ phase dynamics. Thus, our results suggest that in this task the FEF visuomotor module *leads* the rhythmic attention process in a top-down manner, and our model makes the prediction that the good θ phase is associated with the appearance of $\beta_2$ oscillations in FEF *before* γ oscillations in LIP. (Note that, while *Figure 3* appears to support this hypothesis, rigorous testing, even in silico, would require addressing analytical challenges to measuring the precise onset times of rhythmic dynamics.)

Under different stimulation parameters, our model is also able to generate LIP γ oscillations without FEF $\beta_2$ input, provided that the tonic excitation to the LIP granular layers is strong enough. In particular, this means that a very salient stimulus, by providing strong excitation to the LIP input layers, could lead to enhanced sensory processing in a bottom-up fashion. FEF visuomotor cells are able to produce a $\beta_2$ or a γ depending on the strength of inputs from LIP; thus, FEF could participate in different (frequency-specific) functional networks depending on whether visuomotor circuits are activated by highly salient as opposed to near-threshold stimuli.

An important recent rate-based computational model (*Jaramillo et al., 2019*) suggests that pulvinar functions by modulating the gain of feedforward (FF) cortico-cortical connectivity and by altering the balance between feedforward and feedback (FB) cortico-thalamo-cortical (CTC) pathways. This work draws on the fact that the pulvinar interacts with the cortex via two pathways: in addition to feedback pathways in which projections return to the deep layers of the cortical area from which input arrives, pulvinar participates in feedforward pathways in which cortical input originating in one area (A) activates projections to the granular layers of a different cortical area (B), 'boosting' the direct connections from area A to area B. In this previous model, it is the latter cortico-thalamo-cortical (CTC) feedforward pathway through which pulvinar excitability (fixed for the duration of a trial) changes the strength of cortico-cortical connectivity (*Jaramillo et al., 2019*).

In our model, the gain of feedforward connectivity (as indexed by the conductance of synapses from LIP→FEF) plays a similar role in tuning the network's sensitivity for target detection, representing the difference between cued and uncued locations. However, the mechanisms by which pulvinar affects the dynamics of LIP and FEF are quite different. In the current work, pulvinar input has its own temporal dynamics, and induces disparate dynamical responses in FEF and LIP, helping to create a $\beta_2$ rhythm in FEF and a γ rhythm in LIP during the good θ phase. We suggest that these dynamic changes, rather than biasing a competition between FF and FB information transmission, allow the pulvinar to increase the precision of both FF and FB information transmission between FEF and LIP, effectively optimizing the sensitivity of the network to target appearance (at the cued location) during the good θ phase. This increase in the precision of information transmission is due to the deployment

of different rhythmic dynamics in the good and poor θ phases: in the good θ phase, an FEF $\beta_2$ mediates top-down information transmission as well as lowering the threshold for and changing the temporal dynamics of target detection, while an LIP γ mediates bottom-up information transmission; in the poor θ phase, information transmission between FEF and LIP (i.e. spiking in FEF deep layers and LIP superficial layers) occurs at α frequency (which also changes the threshold and temporal dynamics of target detection in FEF). Altering the timescales of rhythmic dynamics of either the FEF $\beta_2$ or the LIP γ decreases the sensitivity of the network. Thus, the dynamics revealed by our model are more complex and more finely tuned to the task at hand than those discussed in the previous model.

This prior work also addressed how pulvinar activity could alter rhythmicity in cortex. In that work, V4 rhythms at lamina-specific frequencies were modeled via E-I networks with lamina-dependent timescales of inhibition, giving rise to a superficial γ and a deep α. Feedback connections arising from the thalamus target interneurons in deep layers (*Cruikshank et al., 2010*; *Zhou et al., 2018*; *Audette et al., 2018*). Thus, in this previous work, pulvinar feedback pathways inhibit deep layer excitatory cells, resulting in less α; and the disinhibition of deep layer E-I networks explains experimental observations that pulvinar lesion results in an increase in α rhythmicity in V4 (*Zhou et al., 2016*). In contrast, in our model, it is top-down β from FEF that disrupts LIP α/$\beta_1$, even though pulvinar input to LIP granular layers also contributes to the disruption of LIP's concatenation $\beta_1$ rhythm. Our model suggests that pulvinar lesion might indirectly increase low-frequency ($\beta_1$) power in LIP, just as it increases α power in V4, although through a different mechanism.

Finally, our results are in agreement with work suggesting that β and γ rhythms transmit top-down and bottom-up information, respectively (*Buschman and Miller, 2007*; *Buffalo et al., 2011*; *van Kerkoerle et al., 2014*; *Bastos et al., 2015*; *van Pelt et al., 2016*). However, they seem to contradict predictive processing accounts in which top-down β rhythms transmit predictions which serve to *diminish* the bottom-up sensory activity they predict (*Bastos et al., 2012*; *van Pelt et al., 2016*), since, in our model, top-down FEF $\beta_2$ rhythmicity *enhances* bottom-up LIP γ rhythmicity. Recent work suggests that such an account of the functions of β and γ rhythms is too simple to account for the full complexity of cortical rhythmic dynamics (*Bastos et al., 2020*). This apparent contradiction may also be resolved by noting that high-level representations (hypotheses) can produce top-down predictions of two types: first-order predictions of the values of given sensory variables (i.e. the content of given representations); and second-order predictions about the relative reliability or informativeness of those variables (regardless of their value *Koelsch et al., 2019*). These second-order predictions about reliability can be used to flexibly weight and combine the multiple sensory inputs that provide evidence relevant to a given hypothesis. For example, while visual and auditory information are both relevant to speech content, visual information may be judged more reliable in a noisy, crowded bar. While first-order predictions may be confirmed (via the canceling out of bottom-up signaling) or may result in prediction errors, second-order predictions may affect the *gain* of certain bottom-up channels, independent of whether those channels produce prediction errors, so that bottom-up information that is (predicted to be) more reliable has more influence on the updating of high-level hypotheses (*Feldman and Friston, 2010*; *Kanai et al., 2015*; *Koelsch et al., 2019*; *Millidge et al., 2021*).

In our model of the cued location, top-down signals from FEF function to increase the gain of LIP superficial layer spiking by disinhibiting the LIP granular layers. We suggest this could be interpreted as a second-order prediction about the relative reliability of bottom-up information from the cued location. Indeed, some accounts of attention identify it more or less explicitly with this gain modulation of bottom-up signals according to their (expected) reliability or precision (*Feldman and Friston, 2010*; *Moran et al., 2013*; *Kanai et al., 2015*). While our conception of attention is broader in general, we note that, in this case, attentional modulation can be interpreted as the gain modulation of bottom-up input based on its expected reliability. Thalamic as well as cortical mechanisms have been implicated in such precision-weighting, with hypotheses singling out pulvinar in particular (*Kanai et al., 2015*); it is tempting to speculate that driving FEF $\beta_2$ rhythms and priming LIP to respond to them may be one concrete way mdPul carries out this function.

## Timescales of observed rhythms optimize task performance

The frequency of FEF and LIP oscillations in our model has consequences for task performance, and the values typically reported in experiments in fact produce peak levels of sensitivity in our model attention network. More precisely, increasing the oscillatory frequency of FEF visuomotor cells from

$\beta_2$ to γ or reducing LIP oscillatory frequency during the good θ phase from γ to $\beta_2$, by changing the time constants of local inhibition, led to higher false alarm rates, with little to no benefit on hit rates.

The optimization of network sensitivity by FEF $\beta_2$ and LIP γ results from the way that these rhythms and their interactions subtly and differentially alter the excitation/inhibition balance in different layers and populations of LIP and FEF during different phases of the θ rhythm (see Task Performance). These observations point to one important role of rhythms in mediating the effects of attention. However, an account of the effects of rhythms in terms of the E/I balance of FEF and LIP neuronal populations is incomplete, as it omits the temporal dynamics of both neural activity and its behavioral consequences. In both experimental data (*Fiebelkorn et al., 2013*; *Fiebelkorn et al., 2019c*) and our model, hit rate (and false alarm rate) varied not only between 'conditions' (i.e. cued vs. uncued object/LIP→FEF connectivity, time scale of inhibition/rhythmic frequency), but also in time. The phase of θ, FEF $\beta_2$, LIP γ, and pulvinar α all have an effect on target detection (and its sensitivity).

We suggest that these temporal dynamics of network activity are not coincidental or epiphenomenological, but play important functional roles. The first of these is to synchronize and sequence precise monitoring and information transmission with (multiple) other computational processes occurring in other brain regions, each of which may have a different characteristic frequency (for example, $\beta_2$ for motor processes, and γ for visual processes). The second is to provide periodic windows of enhanced sensitivity, that enable precise monitoring of a given region of the visual field with acceptable temporal 'coverage', while also minimizing energetic cost and allowing monitoring to be interleaved with other computational imperatives, including monitoring of other regions and motor planning and execution. We further elaborate on this sequencing and segregation of cognitive functions below.

## LIP to FEF connectivity controls the frequency of perceptual sensitivity

Our model replicated experimental results at the behavioral level, with hit rates that varied with the θ phase of pulvinar input to the cortex and which were increased during the good θ phase (i.e. during pulvinar stimulation). In the FEF visual module, VIP cells enabled the detection of low-contrast targets. In our simulations, increasing the LIP→FEF functional connectivity (by increasing the conductance of LIP chemical synapses onto FEF visual cells) increased first hit rates and then false alarm rates, so that intermediate values of connection strength led to peak sensitivity of target detection. These intermediate values of LIP→FEF visual module connectivity were also associated with a strong 8 Hz θ component in the hit rates, which was observed experimentally for targets appearing within the cued (but not the uncued) object (*Fiebelkorn et al., 2013*).

Our model thus suggests that the appearance of a cue on the screen might enhance LIP→FEF visual cell functional connectivity, leading to both increased perceptual sensitivity especially during the poor θ phase and an 8 Hz dominant frequency in the hit rate spectrum. In other words, the 8 Hz hit rates in our model did not come from the division of a 4 Hz rhythm into multiple possible target locations, as previously suggested for similar data (*Landau and Fries, 2012*), but were a consequence of the increased LIP→FEF connection strength for the cued object. In our results, very strong LIP→FEF connectivity led to high false alarm rates, and such excessive connectivity may be linked to hallucinations (as in the detection of a target in the absence of a visual stimulus) or other pathologies in visual perception.

In this work, the increase in LIP→FEF functional connectivity is implemented via an increase in the maximal conductance of the chemical synapses from LIP to FEF. However, there are multiple ways such an increase in drive could arise. For example, either FEF or LIP may be disinhibited by the activation of VIP interneurons in these regions, or thalamus may modulate the gain of LIP→FEF connectivity via a cortico-thalamo-cortical feedforward pathway, as described in the next section. Multiple mechanisms may be implemented in the brain, to allow multiple information streams—e.g., reward statistics from superior colliculus or learned associations from hippocampus or amygdala—to influence the attentional processing of a given visual location.

## Rhythmic sampling and turn-taking between sensory, cognitive, and motor networks

As the good θ phase is associated with improved sensory perception and behavioral performance in attention tasks (*Fiebelkorn and Kastner, 2019b*), one could wonder what need is served by the poor θ phase—and, indeed, the θ rhythm itself—in the first place.

One possibility is that the θ rhythm is a solution to the necessity of attending to multiple objects 'at once', allowing the focus of attention to switch between objects θ-rhythmically. In this account, the poor θ phase for the representation of one object or location would occur simultaneously with the good θ phase for the representation of another object or location (*Fiebelkorn et al., 2013*; *Landau and Fries, 2012*). In our model, this could be achieved with multiple copies of LIP and FEF representing one cortical columnar network for each object to be monitored; pulvinar inputs would select the attended object at any point in time, and manage switching the attentional focus.

Alternatively, the good and poor θ phases may result from an alternation between more and less specific hypotheses about the visual location of target appearance. In this case, the good θ phase would represent preferential processing of visual information from the cued object and location, but the poor θ phase would simply represent a lapse of this preferential processing, and not the preferential processing of the uncued locations (relative to the cued location). This is suggested by the fact that the hit rates at the cued and uncued locations are similar during the (cued location's) poor θ phase (*Fiebelkorn et al., 2013*).

In either account, even when attention is deployed at a single location, the break in attentional focus allowed by the poor θ phase would allow attention and behavior to adapt quickly to changes in visual stimuli and their perceived salience. In this framework, the good θ phase could be described as a sampling phase for one location, while the poor θ phase could be described as an exploration phase (*Fiebelkorn and Kastner, 2019b*). Accounts of binocular rivalry suggest that this alternation between two visual percepts may result from periodic shifts in the levels of confidence ascribed to alternative hypotheses (*Hohwy et al., 2008*), and it is interesting to consider whether similar dynamics apply in the current case. The failure of a target to appear during the good θ phase may be seen as *reducing* confidence in the hypothesis that the target will appear at the cued location. As a result, (relative) confidence may increase in other hypotheses about the location of target appearance (either specific ones related to the uncued locations or general ones that include cued and uncued locations). When the target does not appear during the poor θ phase, confidence in these hypotheses may in turn wane, and the original hypothesis may be 'revived'.

Finally, it has been hypothesized that the poor θ phase could enable the planning and execution of motor actions (*Fiebelkorn and Kastner, 2019b*). Interestingly, in the presence of adequate excitatory drive from mdPul, the FEF visuomotor $\beta_2$ that elicits LIP γ in our model can also be transformed into γ rhythmicity by α/$\beta_1$-frequency LIP input. If the FEF $\beta_2$ rhythm is interpreted as a 'hold signal' (*Engel and Fries, 2010*), pausing FEF γ rhythmicity and consequent attentional shifts, our model suggests that the interactions between FEF and LIP may mediate an oppositional relationship between motor engagement (operationalized by γ in FEF) and sensory processing (operationalized by γ in LIP).

These accounts of the role of the θ rhythm are far from incompatible. The θ rhythm may be a mechanism that *interleaves* periodic assessment of multiple hypotheses about target appearance with motor execution, with movements occurring at the transitions between good and poor θ phases, each of which represents the sampling of evidence for one particular hypothesis. Essentially, movement would simply be the gross motor component of shifts in 'attentional focus', sequenced in turn with preferential processing of multiple salient regions (potentially having different spatial scales) of the visual field.

A deeper understanding of the role of the θ rhythm awaits a detailed investigation of pulvinar activity and the mechanisms of θ rhythmicity and spatial selection in this task. In ongoing work, we are extending the current model to explore how circuits in the pulvinar and the superior colliculus may coordinate ongoing periodic temporal dynamics (as modeled here) between multiple identical networks representing different spatial locations and/or objects, in line with prior learning.

## Modeling choices and limitations

As described in the Introduction, our model represents a number of simplifications relative to the full dynamics of the visual attention network in the context of this task. The model presented here is intended to allow realistic insights into the mechanisms underlying observed brain dynamics and their functional consequences while avoiding excessive complexity. Thus, we have chosen to highlight the cell types and connections necessary to produce the dynamics of interest, while leaving out other cell types and connections. In particular, the spatial structure of the full Egly-Driver task is beyond the scope of the current model. The full task involves spatially informative cues activating subpopulations

that respond selectively to spatial locations and/or objects. This greater mechanistic complexity, in addition to involving a detailed pulvinar model and inputs from other regions, may also involve cell types and connections omitted in the current model, for example VIP cells and excitatory connections between RS cells in LIP superficial layers. VIP cells may participate, for example, by mediating the disinhibition of spatially selective populations. Excitatory synapses among cortical pyramidal cells, which are extremely important for plasticity and formation of cell assemblies, may also be important in the selection of spatially specific subpopulations. However, in our hands, neither are necessary for the current model (a stepping stone to a more complex model of the full Egly-Driver task), in which all the cells in LIP can be thought of as part of a single, cue-relevant cell assembly (although dividing FEF visual neurons into two populations was necessary for the detection of low-contrast targets). The qualitative dynamics of the LIP module remained the same when recurrent connections between RS cells were added ( *Appendix 4—figure 1*) and when VIP cells were added to the superficial layer (as part of a target detection module like the one instantiated in the FEF visual module; *Appendix 4—figure 2*). Similarly, the addition of FS cells in the FEF visuomotor module also did not significantly alter this module's $\beta_2$ dynamics (*Appendix 4—figure 3*).

We have also chosen to model relatively small networks of biophysically detailed neurons. In order for the behavior of small networks to qualitatively match the dynamics of much larger networks, their model parameters often must be different from those of the larger networks. Their dynamics also may be different, in ways that we believe are irrelevant to understanding the qualitative dynamics of the network and the functional implications of interest—for example, rhythms that are more regular than those seen in vivo. Testing the conclusions of models such as ours in larger, more detailed models, as well as in experiments, is essential. However, initial modeling studies in smaller, simplified networks are indispensable in generating initial hypotheses about the mechanisms and functions of brain dynamics.

The parameters of our model are determined in part by the incorporation of model components shown to qualitatively reproduce experimentally observed brain dynamics relevant to other physiological and cognitive contexts: a model of cholinergic attentional modulation in laminar cortex (*Lee and Whittington, 2013*), and models of interlaminar interactions in parietal cortex that instantiate a working memory mechanism (*Kramer et al., 2008*; *Gelastopoulos et al., 2019*). The use of these components imposes additional functional constraints on the parametrization of the model.

Aside from these functional constraints, model parameters are for the most part derived from experimental data in rodents. The differences between rodent and primate neocortex are numerous. Relative to rodents, primates have a 50% greater proportion of interneurons, due in large part to an increase in calretinin, VIP, and neurogliaform interneurons in the superficial layers of association cortex, which distinguishes association and sensory cortices in primates but not rodents (*Džaja et al., 2014*; *Krienen et al., 2020*; *Hodge et al., 2019*). Gene expression patterns of interneuron subtypes also differ more between cortical regions in primates than rodents, suggesting that interneurons may be more specifically tailored to different cortical regions in primates (*Krienen et al., 2020*).

Among these many differences between rodents and primates, there are a few that particularly influence the dynamics of our model. The increase in superficial VIP cells in primates suggests these interneurons are likely part of the circuitry of LIP superficial layers, despite their absence in our model. As discussed above, VIP cells may play a role in determining the allocation of attention between objects in the full Egly-Driver task. However, our results suggest they do not qualitatively alter the dynamics following cue presentation; in our hands, placing a target-detection circuit like the one modeled in the FEF visual module in LIP superficial layers did not alter LIP rhythms in response to low-contrast targets (*Appendix 4—figure 2*). In monkeys relative to rats, PV-positive basket cells have been shown to have a higher input resistance and a lower firing threshold and thus to be more excitable (*Povysheva et al., 2008*). In rats, PV-positive basket cells fired in response to stimulation with a substantial delay, and generated spike trains interrupted by quiescent periods (*Povysheva et al., 2008*), hallmarks of stuttering interneurons exhibiting a D-current (*Golomb et al., 2007*). Although originally parameterized using data from rat hippocampus, our model FS cells lack a D-current, and thus exhibit behavior more similar to primate than to rodent FS cells. Another between-species difference with potential consequences for dynamics in the present model is that supragranular pyramidal neurons are known to exhibit a higher h-current conductance in humans as compared to mice (*Kalmbach et al., 2018*). This higher h-current conductance was shown to promote transfer of θ frequencies from the dendrites to the soma of human pyramidal neurons (*Kalmbach et al., 2018*). In our model, a

higher h-current conductance in superficial pyramidal cells might thus lead to a stronger θ resonance in these cells.

We have chosen to model FEF and LIP as driven θ-rhythmically by simulated mdPul input. As we mention in the Introduction, this is consistent with experimental observations that mdPul exerts a Granger causal influence over FEF and LIP at α frequency only during the good θ phase (*Fiebelkorn et al., 2013*; *Fiebelkorn et al., 2019c*). This α-frequency drive during the good θ phase also amounts to a θ-rhythmic drive from mdPul to FEF and LIP, which is sufficient—in our model—to induce the experimentally observed θ rhythmicity in the dynamics and function of LIP and FEF. However, we cannot rule out the possibility that FEF and LIP also exhibit θ rhythmicity or resonance, or that they (and/or the pulvinar) are paced by θ-rhythmic drive from one or more other brain regions. Indeed, neocortical SOM interneurons have been shown to form electrically coupled networks that generate oscillations at θ frequencies (*Beierlein et al., 2000*; *Fanselow et al., 2008*; *Hu and Agmon, 2015*; *Huang et al., 2020*). These oscillations occur in response to persistent activation by synaptic inputs, metabotropic glutamate agonism, and muscarinic acetylcholine agonism (*Beierlein et al., 2000*; *Fanselow et al., 2008*) as well as in response to salient visual stimuli (*Huang et al., 2020*), and exert a coherent inhibitory influence on cortical pyramidal cells and FS interneurons (*Beierlein et al., 2000*; *Karnani et al., 2016*; *Safari et al., 2017*). Coupled with a potential θ resonance in superficial RS cells induced by increased levels of h-current in humans (see above), such a θ-rhythmic SOM network could result in endogenous θ rhythmicity in LIP superficial layers. Whether and how this θ rhythmicity would change our results is an important question for future investigations.

Finally, in our FEF visual module, the interactions of inhibitory PV, SOM, and VIP interneurons within and across cortical columns produce a circuit that is particularly sensitive to low-contrast stimuli modeled as a weak input to one column and not another. The functional role of interactions between these three interneuron types has received considerable attention (*Karnani et al., 2016*; *Yang et al., 2016*; *Lee et al., 2017*; *Dipoppa et al., 2018*; *Krabbe et al., 2019*; *Lee et al., 2019*; *Garrett et al., 2020*; *Hertäg and Sprekeler, 2020*; *Keller et al., 2020*; *Millman et al., 2020*; *Veit et al., 2021*). This literature highlights the role of VIP interneurons in disinhibiting pyramidal cells (via inhibition of SOM cells) to produce context-dependent stimulus sensitivity. VIP-mediated disinhibition has been implicated in (i) detection of stimulus and surround differences (*Keller et al., 2020*), (ii) the creation of a high gain regime that increases sensitivity to weak inputs (*Millman et al., 2020*), (iii) the desynchronization of gamma oscillations representing spatially discontiguous visual features (*Veit et al., 2021*), (iv) the facilitation of auditory cortical stimulus responses by nicotinic acetylcholine receptor activation (*Askew et al., 2019*), (v) response reversal (*Garcia Del Molino et al., 2017*), (vi) the coding of prediction errors (*Hertäg and Sprekeler, 2020*), and (vii) the gating of information flow between cortical regions (*Yang et al., 2016*; *Lee et al., 2017*; *Krabbe et al., 2019*; *Lee et al., 2019*). These results motivate in part the circuitry of the FEF visual module in the present model. A number of other models have been constructed to explore these effects, including rate models (*Millman et al., 2020*; *Dipoppa et al., 2018*; *Hertäg and Sprekeler, 2020*), mean-field models (*Veit et al., 2021*), and spiking models (*Yang et al., 2016*; *Lee et al., 2017*; *Garcia Del Molino et al., 2017*; *Veit et al., 2021*). While a full understanding of the relationships between these models is beyond the scope of the current work, in our model, both pyramidal disinhibition via VIP-mediated SOM suppression and intercolumnar inhibition via SOM cells were required for the amplification of small inputs. The latter feature is not common in existing models of VIP-SOM interactions (but see *Veit et al., 2021*). Another requirement is that our VIP interneurons exhibit a steeper f-I curve than our SOM interneurons (*Appendix 1—figure 1*). VIP interneurons are among the interneurons with the highest input resistance and are therefore among the most excitable (*Rudy et al., 2011*), and cortical and thalamic inputs are greatest onto VIP interneurons as opposed to PV or SOM cells (*Wall et al., 2016*). Our model is thus in line with recent investigations suggesting that a stabilized supralinear network with three interneuron types having a distribution of f-I curves can mediate an input-strength dependent switch between a (VIP-mediated) high gain regime that increases sensitivity to weak inputs and a (SOM-mediated) low gain regime that stabilizes the network response to strong inputs (*Millman et al., 2020*). Such disinhibitory mechanisms are clearly widespread in the cortex and an important subject for future work.

## Methods

### Equations governing model neurons

The model presented in this article was adapted from *Kramer et al., 2008* and *Gelastopoulos et al., 2019*. These laminar models reproduce somatosensory and parietal rhythmic dynamics, and the cells are reductions of detailed multi-compartment representations of rat cortical neurons (*Traub et al., 2005*). The current model consisted of Hodgkin-Huxley neurons, modeled with a single compartment except for LIP intrinsically bursting (IB) cells which were modeled with four compartments. The membrane potential $V$ of each single-compartment neuron and each IB cell compartment followed an equation of the form:

$$C\frac{dV}{dt} = -\sum I_{\mathrm{m}} - \sum I_{\mathrm{syn}} - I_{\mathrm{ran}}$$

where $C$ is the membrane capacitance, $I_{\mathrm{m}}$ are the membrane currents, $I_{\mathrm{syn}}$ are the synaptic currents, and $I_{\mathrm{ran}}$ is a random input current. Some cells are also electrically coupled with gap junctions. The membrane capacitance was equal to $0.9 \mu F/cm^2$ for all neurons except VIP interneurons, for which it was $2 \mu F/cm^2$.

Our regular spiking (RS) pyramidal cell model included a leak current $I_{\mathrm{L}}$, a sodium current $I_{\mathrm{Na}}$, a potassium current $I_{\mathrm{K}}$, and a hyperpolarization activated current (h-current) $I_{\mathrm{AR}}$. The conductances of these currents (*Table 1*) were adapted in *Traub et al., 2005* from experiments on rat hippocampal neurons (*Martina and Jonas, 1997*). There were 80 RS cells in our LIP superficial layer model, 20 in LIP granular layer, 20 in visuomotor FEF, and 20 in visual FEF. The 20 decision cells in visuomotor FEF were also RS cells.

Our fast spiking (FS) interneuron model in LIP and FEF included a leak current $I_{\mathrm{L}}$, a sodium current $I_{\mathrm{Na}}$, and a potassium current $I_{\mathrm{K}}$. The conductances of these currents (*Table 2*) were adapted in *Traub et al., 2005* from experimental studies of rat hippocampal neurons ($I_{\mathrm{L}}$, $I_{\mathrm{Na}}$, & $I_{\mathrm{K}}$; *Martina and Jonas, 1997*; *Martina et al., 1998*). There were 20 FS cells each in the LIP superficial layer, LIP granular layer and FEF visual module.

Our SOM cell model included a leak current $I_{\mathrm{L}}$, a sodium current $I_{\mathrm{Na}}$, a potassium current $I_{\mathrm{K}}$, and an h-current $I_{\mathrm{AR}}$. The conductance for $I_{\mathrm{AR}}$ was modified from experiments on guinea pig thalamic interneurons *Huguenard and McCormick, 1992*; all other conductances were the same as for FS cells (*Table 3*). There were 20 SOM cells each in the LIP superficial layer, LIP deep layer, and FEF visuomotor and visual modules.

Our VIP cell model included a leak current $I_{\mathrm{L}}$, a sodium current $I_{\mathrm{Na}}$, a potassium current $I_{\mathrm{K}}$ and a potassium D-current $I_{\mathrm{D}}$ with conductances taken from studies of guinea pig cortical neurons (*Golomb et al., 2007*, *Table 4*). There were 20 VIP cells in the FEF visual module.

Our IB cell model consisted of a somatic compartment, an axonal compartment, and apical and basal dendritic compartments, with electrical coupling between them (*Table 5*). All compartments included a leak current $I_{\mathrm{L}}$, a sodium current $I_{\mathrm{Na}}$, and a potassium current $I_{\mathrm{K}}$, whose kinetics were

**Table 1.** Ion channel parameters for RS cells.

| Channel | $g$ $(mS/cm^2)$ | $E$ $(mV)$ | $p$ | $q$ | $\tau_m$(ms) | $m_\infty$ | $\tau_h$(ms) | $h_\infty$ |
|---|---|---|---|---|---|---|---|---|
| L | 1 | –70 | 0 | 0 | - | - | - | - |
| Na | 200 | 50 | 3 | 1 | - | $\frac{1}{1+e^{\frac{-V-34.5}{10}}}$ | $0.15 + \frac{1.15}{1+e^{\frac{V+33.5}{15}}}$ | $\frac{1}{1+e^{\frac{V+59.4}{10.7}}}$ |
| K | 20 | –95 | 4 | 0 | $0.25 + 4.35 \cdot e^{-\frac{|V+10|}{10}}$ | $\frac{1}{1+e^{\frac{-V-29.5}{10}}}$ | - | - |
| h | 25 | –35 | 1 | 0 | $\frac{1}{e^{-14.6-0.086V}+e^{-1.87+0.07V}}$ | $\frac{1}{1+e^{\frac{V+87.5}{5.5}}}$ | - | - |

**Table 2.** Ion channel parameters for FS cells.

| Channel | $g$ $(mS/cm^2)$ | $E$ $(mV)$ | $p$ | $q$ | $\tau_m$ | $m_\infty$ | $\tau_h$ | $h_\infty$ |
|---|---|---|---|---|---|---|---|---|
| L | 1 | –65 | 0 | 0 | - | - | - | - |
| Na | 200 | 50 | 3 | 1 | - | $\frac{1}{1+e^{\frac{-V-38}{10}}}$ | $0.225 + \frac{1.125}{1+e^{\frac{V+37}{15}}}$ | $\frac{1}{1+e^{\frac{V+58.3}{6.7}}}$ |
| K | 20 | –100 | 4 | 0 | $0.25 + 4.35e^{-\frac{|V+10|}{10}}$ | $\frac{1}{1+e^{\frac{-V-27}{11.5}}}$ | - | - |

**Table 3.** Ion channel parameters for SOM cells.

| Channel | $g$ $(mS/cm^2)$ | $E$ $(mV)$ | $p$ | $q$ | $\tau_m$ | $m_\infty$ | $\tau_h$ | $h_\infty$ |
|---|---|---|---|---|---|---|---|---|
| L | 6 | –65 | 0 | 0 | - | - | - | - |
| Na | 200 | 50 | 3 | 1 | - | $\frac{1}{1+e^{\frac{-V-38}{10}}}$ | $0.225 + \frac{1.125}{1+e^{\frac{V+37}{15}}}$ | $\frac{1}{1+e^{\frac{V+58.3}{6.7}}}$ |
| K | 10 | –100 | 4 | 0 | $0.25 + 4.35 \cdot e^{-\frac{|V+10|}{10}}$ | $\frac{1}{1+e^{\frac{-V-27}{11.5}}}$ | - | - |
| h | 50 | –35 | 1 | 0 | $\frac{1}{e^{-14.6-0.086V}+e^{-1.87+0.07V}}$ | $\frac{1}{1+e^{\frac{V+75*mV}{5.5}}}$ | - | - |

taken from experiments on rat hippocampal neurons (*Martina and Jonas, 1997*). The axonal compartment and both dendritic compartments also included a potassium M current ($I_{KM}$). Additionally, the dendritic compartments included an h-current $I_{AR}$ (based originally on experiments on guinea pig thalamic interneurons; *Huguenard and McCormick, 1992*) and a high-threshold calcium current $I_{CaH}$ (parameters based on studies of rat thalamocortical neurons; *Destexhe et al., 1998*). There were 20 IB cells in the LIP module.

Each membrane channel's current $I$ was expressed in the form.

$$I = g.m^p.h^q.(V - E)$$

where $g$ is the channel conductance, $E$ its reversal potential, and $m$ and $h$ are the gating variables, which followed the differential equation.

$$\frac{dx}{dt} = \frac{1}{\tau_x} \cdot (x_\infty - x), \quad x \in \{m, h\}$$

or, in an equivalent manner $\frac{dx}{dt} = \alpha_x(V) \cdot (1 - x) - \beta_x(V) \cdot x, \quad x \in \{m, h\}$

with the exception of the sodium currents for which $m = m_\infty(V)$. The expressions and values for the variables and parameters $g$, $E$, $p$, $q$, $\tau_m$ and $m_\infty$ (or $\alpha_m$ and $\beta_m$), $\tau_h$ and $h_\infty$ (or $\alpha_h$ and $\beta_h$) for each channel and each cell type are detailed in *Tables 1–5*.

The random current $I_{ran}$ received by each cell followed the equation:

$$I_{ran} = \sigma_{ran}^2 \cdot dW$$

where $dW$ is Gaussian white noise of unit variance per $ms$, and $\sigma_{ran}^2$ is the intensity of this additive white noise. The value of the parameter $\sigma_{ran}^2$ used for each cell type is given in *Table 6*. An additional

**Table 4.** Ion channel parameters for VIP cells.

| Channel | g (mS/cm²) | E (mV) | p | q | $\tau_m$ | $m_\infty$ | $\tau_h$ | $h_\infty$ |
|---|---|---|---|---|---|---|---|---|
| L | 0.25 | −70 | 0 | 0 | - | - | - | - |
| Na | 112.5 | 50 | 3 | 1 | - | $\frac{1}{1+e^{-\frac{V+24}{11.5}}}$ | $0.5 + \frac{14}{1+e^{\frac{V+60}{12}}}$ | $\frac{1}{1+e^{\frac{V+58.3}{6.7}}}$ |
| K | 225 | −90 | 2 | 0 | $\left(0.087 + \frac{11.4}{1+e^{\frac{V+14.6}{8.8}}}\right) \cdot \left(0.087 + \frac{11.4}{1+e^{\frac{V-1.3}{18.7}}}\right)$ | $\frac{1}{1+e^{-\frac{V+12.4}{6.8}}}$ | - | - |
| D | 4 | −90 | 3 | 1 | 2 | $\frac{1}{1+e^{-\frac{V+50}{20}}}$ | 150 | $\frac{1}{1+e^{\frac{V+70}{6}}}$ |

random noise of amplitude $30\mu A/cm^2$ was provided to the FEF visuomotor module in the poor theta phase, representing arrythmic inputs from the pulvinar.

## Network connectivity

The neurons in our model interacted with each other through AMPA and GABA$_A$ synapses as well as gap junctions. The synaptic current $I_{syn}$ received by each cell (or compartment) was the sum of the synaptic currents $I_{syn,i}$, one for each incoming synapse. The contribution of each synapse was given by the equation.

**Table 5.** Ion channel parameters for IB cells.

| Channel | g (mS/cm²) | E (mV) | p | q | $\tau_m$ | $m_\infty$ | $\tau_h$ | $h_\infty$ |
|---|---|---|---|---|---|---|---|---|
| L | 2 (a.d., b.d.)<br>1 (soma)<br>0.25 (axon) | −70 | 0 | 0 | - | - | - | - |
| Na | 125 (a.d., b.d.)<br>50 (soma)<br>100 (axon) | 50 | 3 | 1 | - | $\frac{1}{1+e^{\frac{-V-34.5}{10}}}$ | $0.15 + \frac{1.15}{1+e^{\frac{V+33.5}{15}}}$ | $\frac{1}{1+e^{\frac{V+59.4}{10.7}}}$ |
| K | 10 (a.d., b.d., soma)<br>5 (axon) | −95 | 4 | 0 | $0.25 + 4.35 \cdot e^{-\frac{|V+10|}{10}}$ | $\frac{1}{1+e^{\frac{-V-29.5}{10}}}$ | - | - |
| h | 155 (a.d.)<br>115 (b.d) | −25 | 1 | 0 | $\frac{1}{e^{-14.6-0.086V}+e^{-1.87+0.07V}}$ | $\frac{1}{1+e^{\frac{V+75}{5.5}}}$ | - | - |

| Channel | g (mS/cm²) | E (mV) | p | q | $\alpha_m$ | $\beta_m$ | $\alpha_h$ | $\beta_h$ |
|---|---|---|---|---|---|---|---|---|
| M | 0.75 (a.d., b.d)<br>1.5 (axon) | −95 | 1 | 0 | $\frac{0.02}{1+e^{\frac{-V-20}{5}}}$ | $0.01 \cdot e^{\frac{-V-43}{18}}$ | - | - |
| CaH | 6.5 (a.d, b.d) | 125 | 2 | 0 | $\frac{1.6}{1+e^{-0.072\cdot(-V-5)}}$ | $0.02 \cdot \frac{V+8.9}{e^{\frac{V+8.9}{5}}-1}$ | - | - |

**Table 6.** Random input amplitude for each cell type.

| Cell Type | Noise Amplitude $\sigma^2_{\mathrm{ran}}(mA/cm^2)$ |
|---|---|
| RS | 0.075 |
| FS | 0.025 |
| SOM | 0.025 |
| VIP | 0 |
| IB (axon) | 0.0125 |
| IB (soma) | 0 |
| IB (apical dendrites) | 0.0025 |
| IB (basal dendrites) | 0.0025 |

$$I_{\mathrm{syn},i} = s_i \cdot g_i \cdot (V - E_i)$$

where $g_i$ is the maximum conductance of the synapse and $E_i$ is its reversal potential. The synaptic gating variable $s_i$ depended on the rise and decay time constants of the synapse $\tau_{\mathrm{r},i}$ and $\tau_{\mathrm{d},i}$ as well as the membrane potential of the presynaptic neuron $V_{\mathrm{pre},i}$, according to the equation.

$$s_i = -\frac{s_i}{\tau_{d,i}} + \frac{1-s_i}{\tau_{r,i}} \cdot 0.5 \cdot \left(1 + tanh\frac{V_{pre,\,i}}{10}\right)$$

The parameters chosen for each synapse type in the network are summed up in **Table 7** (for the reversal potentials, rise and decay time constants) and **Table 8** (for the maximum conductances).

Similarly, the total gap junction current received by a cell $I_{\mathrm{gap}}$ was the sum of all contributions $I_{\mathrm{gap},i}$ received from each individual cell or compartment contacting it, described by the equation.

$$I_{\mathrm{gap},i} = g_i \cdot (V - V_{\mathrm{pre},i})$$

where $g_i$ is the maximum conductance of the gap junction and $V_{\mathrm{pre},i}$ is the membrane potential of the other neuron or compartment involved in the gap junction. Gap junctions were present between LIP superficial RS cells with a maximum conductance $g_i = 0.04$, between LIP superficial SOM cells with a maximum conductance $g_i = 0.2$, between LIP IB cell axons with a maximum conductance $g_i = 0.0025$, and between the different compartments of each LIP IB cell, with conductances summed up in **Table 9**. Of note, the maximal conductances of these gap junctions were likely larger than what would be measured for single pairs of cells in vivo, as is appropriate for currents in a smaller network.

## Inputs

Some cell populations in the network received an external input current $I_{\mathrm{inp}}$, following the set of equations.

$$I_{\mathrm{inp}} = s_{\mathrm{inp}} \cdot g_{\mathrm{inp}} \cdot (V - E_{\mathrm{inp}})$$
$$\frac{ds_{\mathrm{inp}}}{dt} = -\frac{s_{\mathrm{inp}}}{\tau_{d,inp}} + \frac{(1-s_{\mathrm{inp}})}{\tau_{r,inp}} \cdot 0.5 \cdot \left(1 + \tanh\frac{V_{\mathrm{inp}}}{10}\right)$$

where $g_{\mathrm{inp}}$ is the maximum conductance of the input synapse and $E_{\mathrm{inp}} = 0mV$ is its reversal potential. The synaptic gating variable $s_{\mathrm{inp}}$ depended on the rise and decay time constants of the synapse $\tau_{\mathrm{r},inp} = 0.1ms$ and $\tau_{\mathrm{d},inp} = 0.5ms$ as well as the membrane potential of the presynaptic neuron $V_{\mathrm{inp}}$.

Some cell populations also received input from neuronal populations that were not modeled explicitly (i.e. with full Hodgkin-Huxley dynamics). Each neuron in these input populations was modeled as a periodic spike train with frequency $f$, having inter-spike intervals uniformly randomly distributed in the interval $(0.9f, 1.1f)$, and independently chosen for each input neuron. The efferent synapses of these input populations followed the equations

$$\frac{dV_{\mathrm{inp}}}{dt} = \frac{1}{\tau_{\mathrm{inp}}} \cdot (V_{\mathrm{low}} - V_{\mathrm{inp}})$$
$$\text{on spike}: \quad V_{\mathrm{inp}} \leftarrow V_{\mathrm{high}}$$

that is, whenever a spike occurred, the membrane potentials of their synaptic targets were reset to $V_{\mathrm{inp}} = V_{\mathrm{high}} = 0mV$ before

**Table 7.** Synaptic reversal potentials and time constants for each synapse type in the network.

| Synapse type | Source neuron | $E_i$ | $\tau_{\mathrm{r},i}$ | $\tau_{\mathrm{d},i}$ |
|---|---|---|---|---|
| AMPA | RS, IB | 0 mV | 0.125ms | 1ms |
| NMDA | RS | 0 mV | 12.5ms | 125ms |
| GABA (fast inhibition) | FS | –80 mV* | 0.25ms | 5ms |
| GABA (slow inhibition) | SOM, VIP | –80 mV | 0.25ms | 20ms |

*FS to FS synapses have a –75 mV reversal potential, instead of –80 mV.

**Table 8.** Maximum conductances of the synapses in the network (in $mS/cm^2$).

| Source region | | LIP | | | | | | | FEF vm | | FEF v | | | |
|---|---|---|---|---|---|---|---|---|---|---|---|---|---|---|
| Neuron | | RS | FS | SOM | g.RS | g.FS | IB | d.SOM | RS | SOM | RS | FS | SOM | VIP |
| Target region | Neuron | | | | | | | | | | | | | |
| LIP | RS | -0.0250 | 6.25 | 2 | 2 | 0.1 | - | - | - | - | - | - | - | - |
| | FS | .225 | 2 | 0.2 | 0.1 | - | 0.08 | - | - | - | - | - | - | - |
| | SOM | - | 0.4 | 7 | - | - | 0.045 | - | - | - | - | - | - | - |
| | g.RS | - | - | - | 0.5 | 1 | - | - | - | - | - | - | - | - |
| | g.FS | 1 | - | - | 1 | 0.3 | - | 1 | - | - | - | - | - | - |
| | IB | /60* | - | 0.4 | - | - | 1/500 | 10 | - | - | - | - | - | - |
| | d.SOM | - | - | - | - | - | - | - | 0.05 | - | - | - | - | - |
| FEF vm | | - | - | - | - | - | - | - | 0.6 | 0.8 | - | - | - | - |
| | RS | 0.009 | - | - | - | - | - | - | 0.5 | - | - | - | - | - |
| | SOM | 0.009 | - | - | - | - | - | - | - | - | - | - | - | - |
| FEF v | RS | 0.015 | - | - | - | - | - | - | - | - | 0.2 | 0.2 | 1 | - |
| | FS | - | - | - | - | - | - | - | - | - | 0.2 | 0.2 | - | - |
| | SOM | 0.025 | - | - | - | - | - | - | - | - | - | - | - | 0.7 |
| | VIP | 0.005 | - | - | - | - | - | - | - | - | - | - | 0.01 | - |
| FEF m | RS | - | - | - | - | - | - | - | 0.1† | - | 0.08‡ | - | - | - |

*For AMPA synapse. For NMDA : $g_i$ = 1/240.

†For AMPA synapse. For NMDA : $g_i$ = 0.01.

‡*For NMDA synapse. For AMPA : $g_i$ = 0.

exponentially decaying to their baseline value $V_{inp} = V_{low} = -80mV$ with a decay time constant of $\tau_{inp} = 0.5ms$. To implement the sinusoidal or sawtooth shape of the θ rhythm, we associated the two volleys of mdPul input occurring during each good θ phase with different maximal conductances $g_{inp}$. The frequency $f$ and conductance $g_{inp}$ for each input synapse are given in *Table 10*.

## Simulations

All the simulations were performed using the Brian2 libraries for Python (*Stimberg et al., 2014*), using a 4th order Runge-Kutta method to solve the differential equations defining the model, and a 0.01ms timestep. All simulations were 2 s long, and the first 200 ms of each simulation was discarded to remove transients. Targets were presented at cue-target delays spaced every 50ms from 200ms to 1800ms. An ensemble of 50 simulations was computed for each delay value, with randomized noise and randomized temporal jitter in the inputs. All code is available at https://github.com/benpolletta/egly-driver-network (copy archived at *Pittman-Polletta, 2023*).

**Table 9.** Conductances of gap junctions between IB cell compartments.

| Source compartment | Target Compartment | |
|---|---|---|
| Soma | Apical Dendrite | 0.2 |
| Soma | Basal Dendrite | 0.2 |
| Soma | Axon | 0.3 |
| Apical Dendrite | Soma | 0.4 |
| Basal Dendrite | Soma | 0.4 |
| Axon Dendrite | Soma | 0.3 |

## Statistics and data analysis

All analyses were performed in Python, with the exception of spectrograms in *Figure 3C*, *Figure 4C* which were computed in Matlab.

The statistics of target detection were computed for 50 simulations at each of 32 cue-target intervals (see above). A 'hit' was recorded if more than half of FEF decision cells spiked at least once during the $100ms$ -long window in which the target was presented. A 'false alarm' was recorded if more than half of FEF decision

**Table 10.** Nominal frequency and maximum conductances (in $mS/cm^2$) of inputs to the model, coming from mdPul, LIP, FEF or V4.
The two conductance values given for mdPul inputs are for the first and second input volleys occurring during each good θ phase.

| Parameter | mdPul input | LIP input (only when FEF is modeled alone) | FEF input (only when LIP is modeled alone) | V4 input (only during target appearance) |
|---|---|---|---|---|
| Good theta $f$ | 13 Hz | 50 Hz | 25 Hz | 50 Hz |
| Poor theta $f$ | - | 13 Hz | - | 50 Hz |
| $g_{inp}$ | to LIP gran. RS: 2.5/5<br>to LIP gran. FS: 2.5/5<br>to FEF vm RS: 2.5/5<br>to FEF vm SOM: 3 | to FEF vm RS : 3<br>to FEF vm SOM : 3<br>to FEF v VIP : 2.5<br>to FEF v SOM : 7.5<br>to FEF v RS : 7.5 | to LIP deep SOM : 5 | to FEF v VIP : 3<br>to FEF v SOM : 2.5 |

cells spiked within a $25ms$-long window in which no target was presented. Given a hit rate $H$ and a false alarm rate $F$, the sensitivity was calculated as

$$D' = z^{-1}(1 - F) - z^{-1}(1 - H),$$

where $z^{-1}(r)$ is defined as the inverse cumulative distribution function for the standard normal distribution on the interval $(0, 1)$ (i.e., $z^{-1}(r)$ is the real number whose standard normal $p$-value is $r$), $z^{-1}(0) = z^{-1}(\epsilon)$, and $z^{-1}(1) = z^{-1}(1 - \epsilon)$ for $\epsilon = 0.1$.

The θ phase was determined from synthetic mdPul α inputs, which had a period of 250ms and a duty cycle of 125ms. Thus, 0° occurred at the onset of the mdPul α, and 180° occurred at the offset of mdPul α. The LIP $\beta_1$ phase during the poor θ phase was determined from the timing of superficial RS cells spiking (RS cells spiking was considered to be at 0°), assuming a $\beta_1$ frequency of 10 Hz.

LIP LFPs were computed by averaging the membrane potentials of superficial RS cells; FEF visuo-motor LFPs were computed by averaging the membrane potentials of visuomotor RS cells.

Power spectra were calculated using SciPy's *signal.periodogram* with a flat top window. The spec-trograms in *Figure 3C* and *Figure 4C* were calculated from LFPs taken from a single 2 s simulation, via convolution with complex Morlet wavelets (obtained with Matlab's *dftfilt3*) with frequencies between 9 and 60 Hz; the width of these wavelets was linearly interpolated from 4 cycles at 9 Hz to 12 cycles at 60 Hz.

## Additional information

### Funding

| Funder | Grant reference number | Author |
|---|---|---|
| National Institutes of Health | P50 MH109429 | Ian C Fiebelkorn<br>Nancy J Kopell<br>Sabine Kastner |
| National Institute of Mental Health | RO1-MH64043 | Ian C Fiebelkorn<br>Sabine Kastner |
| National Eye Institute | RO1-EY017699 | Ian C Fiebelkorn<br>Sabine Kastner |

| Funder | Grant reference number | Author |
|--------|------------------------|--------|

The funders had no role in study design, data collection and interpretation, or the decision to submit the work for publication.

## Author contributions

Amélie Aussel, Conceptualization, Software, Formal analysis, Validation, Investigation, Visualization, Methodology, Writing – original draft, Writing – review and editing; Ian C Fiebelkorn, Conceptualization, Writing – original draft, Writing – review and editing; Sabine Kastner, Conceptualization, Funding acquisition, Visualization, Writing – original draft, Project administration, Writing – review and editing; Nancy J Kopell, Conceptualization, Resources, Supervision, Funding acquisition, Investigation, Visualization, Methodology, Writing – original draft, Project administration, Writing – review and editing; Benjamin Rafael Pittman-Polletta, Conceptualization, Supervision, Validation, Investigation, Visualization, Methodology, Writing – original draft, Writing – review and editing

## Author ORCIDs

Amélie Aussel http://orcid.org/0000-0003-0498-2905
Sabine Kastner http://orcid.org/0000-0002-9742-965X
Benjamin Rafael Pittman-Polletta http://orcid.org/0000-0002-6798-7191

## Decision letter and Author response

Decision letter https://doi.org/10.7554/eLife.67684.sa1
Author response https://doi.org/10.7554/eLife.67684.sa2

---

# Additional files

## Supplementary files

• MDAR checklist

## Data availability

The current manuscript is a computational study, so no data have been generated for this manuscript. Modelling code is available on the ModelDB open repositories.

The following dataset was generated:

| Author(s) | Year | Dataset title | Dataset URL | Database and Identifier |
|-----------|------|---------------|-------------|--------------------------|
| Aussel A | 2023 | LIP and FEF rhythmic attention model | https://senselab.med.yale.edu/modeldb/enterCode?model=267619 | ModelDB, 267619 |

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

# Appendix 1

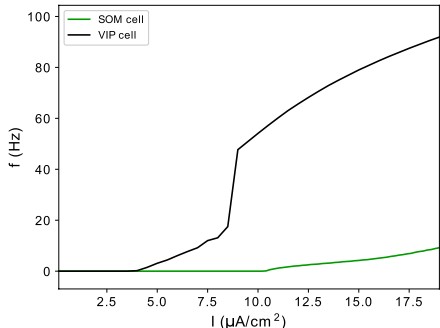

**Appendix 1—figure 1.** f-I curves of model SOM and VIP cells.

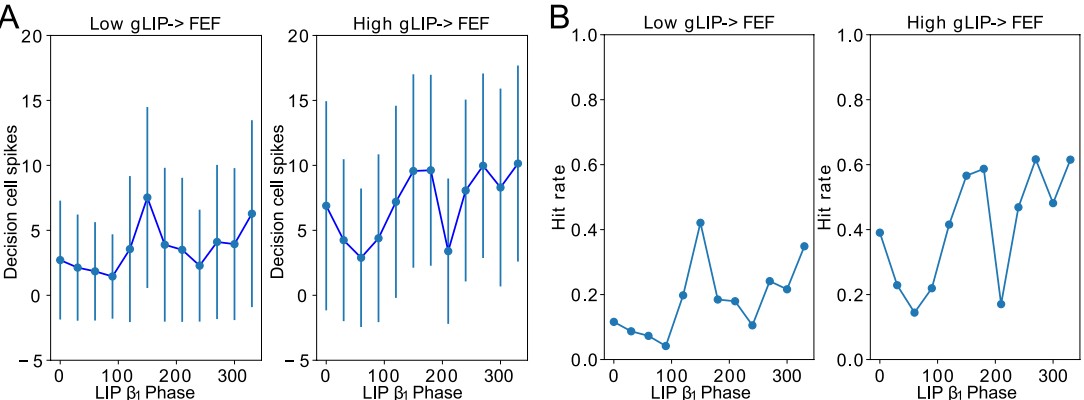

**Appendix 1—figure 2.** Conductance of LIP → FEF synapses increases dependence of decision cell spiking and hit rate on LIP $\beta_1$ phase. (**A**). Decision cell spiking (mean±SD) as a function of LIP $\beta_1$ phase (see Methods). (**B**). Hit rate as a function of LIP $\beta_1$ phase.

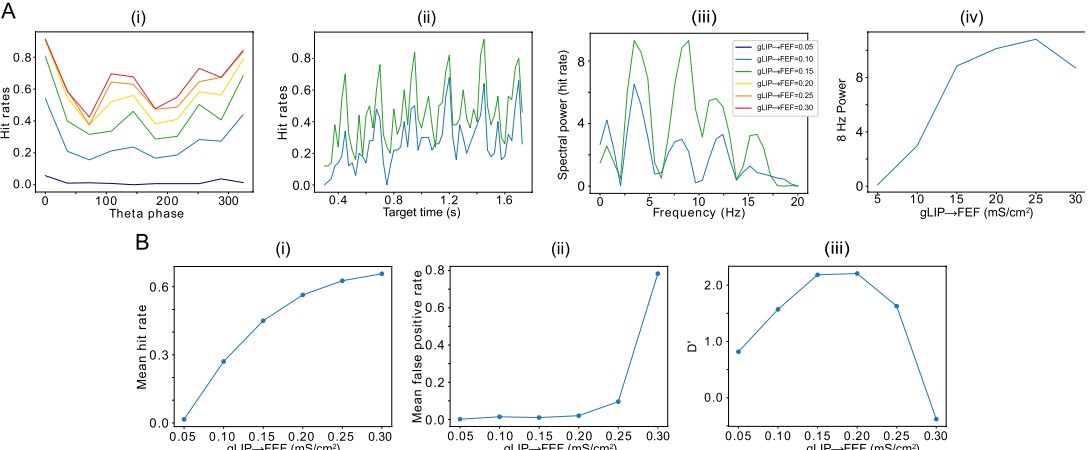

**Appendix 1—figure 3.** The power of 8 Hz rhythmicity in hit-rate depends on mdPul input increasing across the good $\theta$ phase. Simulation results as in **Figure 7** are shown for a network with mdPul input of constant maximal conductance. (**A**). Increased $g_{\text{LIP}\to\text{FEFv}}$ leads to a stronger 8Hz component in the hit rates. (**i**) Hit rate as a function of $\theta$ phase plotted for values of $g_{\text{LIP}\to\text{FEFv}}$ between 0.05 and 0.3 (see (**iii**) for legend). (**ii**) Hit rate as a function of the cue-target interval for $g_{\text{LIP}\to\text{FEFv}} = 0.10 mS/cm^3$ and $g_{\text{LIP}\to\text{FEFv}} = 0.15 mS/cm^3$ (**iii**) Power spectra of the hit
*Appendix 1—figure 3 continued on next page*

*Appendix 1—figure 3 continued*

rate time series shown in (**ii**). (**iv**) 8 Hz power as a function of $g_{LIP \to FEFv}$ (**B**). Network sensitivity as a function of LIP $\to$ FEF connectivity. (i) Hit rate as a function of $g_{LIP \to FEFv}$ (**ii**) False alarm rate as a function of $g_{LIP \to FEFv}$ (**iii**) D' as a function of $g_{LIP \to FEF}$.

# Appendix 2

## Influence of Tonic Excitation on LIP Behavior

The tonic excitation range for the LIP module to exhibit behavior matching experimental observations is rather broad (between -12.5 and 2.5 for FS cells and below 17.5 for RS cells), but choosing excitation levels out of this range can disturb both the rhythm during the poor θ phase and the γ rhythm during the good θ phase. Excessive tonic excitation of granular FS cells (which inhibit superficial RS cells; *Figure 3A*) prevents superficial RS cell spiking and thus disturbs the poor θ phase rhythm (*Appendix 2—figure 1A*). Insufficient tonic drive to granular FS cells disturbs the rhythm by allowing granular RS cells to spike and entrain the superficial layer (*Appendix 2—figure 1B*). Finally, insufficient tonic excitation to granular RS cells disturbs the good θ phase γ rhythm, rendering RS cells unable to sustain their activity outside the windows of disinhibition provided by deep SOM cell spiking (*Appendix 2—figure 1C*).

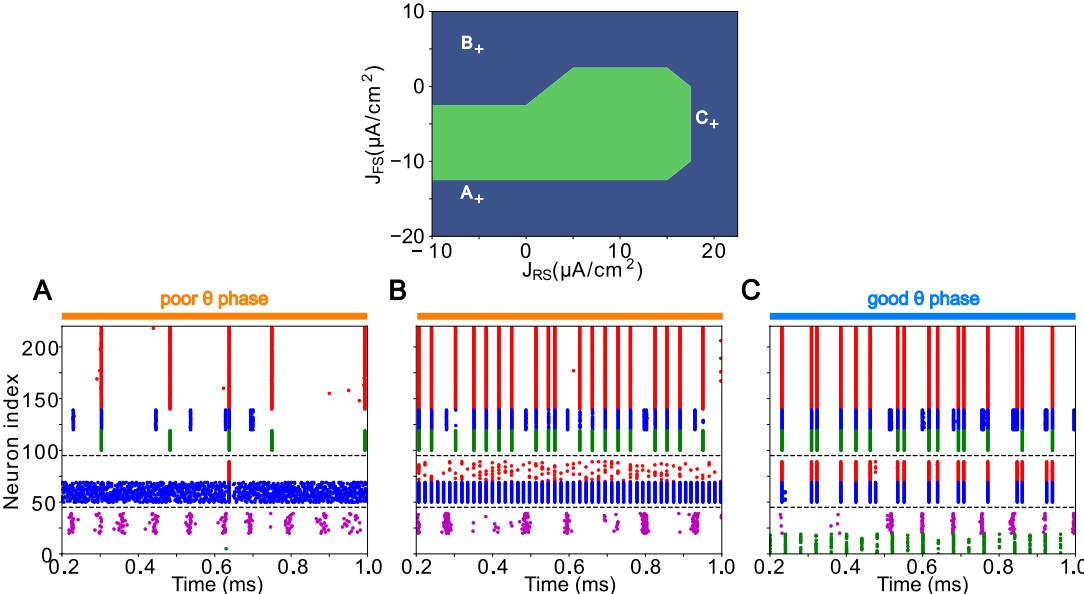

**Appendix 2—figure 1.** LIP good and poor $\theta$ phase behavior depends on input layer tonic excitation. Top: Appropriate tonic excitation range needed to obtain a γ rhythm in the good $\theta$ phase and $\beta_1$ rhythm in the poor $\theta$ phase. The crosses indicate the sets of values used in simulations for (**A, B, and C**) below. (**A**). Excessive tonic excitation to input layer FS cells disturbs the poor $\theta$ phase $\beta_1$ rhythm by preventing superficial RS cells spikes. (**B**). Insufficient tonic excitation to input layer FS cells disturbs the poor $\theta$ phase $\beta_1$ rhythm by allowing input RS cells to spike and entrain the superficial layer. (**C**). Insufficient tonic excitation to input layer RS cells disturbs the good $\theta$ phase γ rhythm.

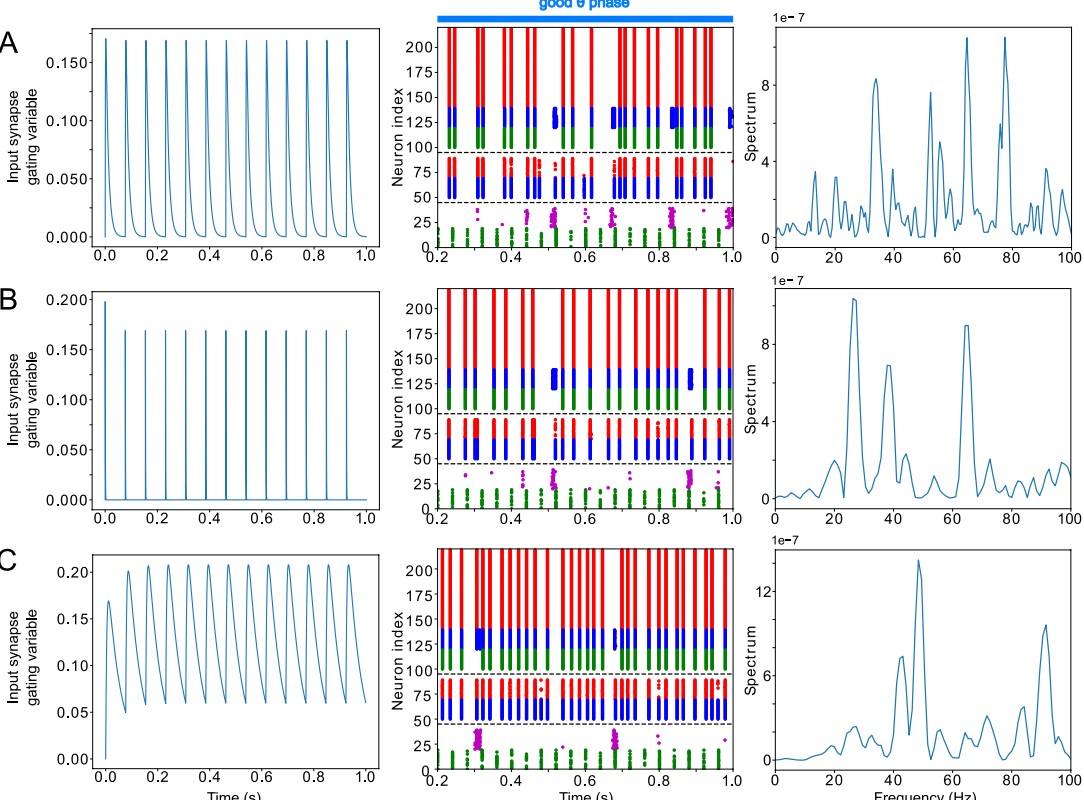

**Appendix 2—figure 2.** LIP good $\theta$ phase γ can be obtained with mdPul α stimulation of various widths. (**A**). LIP good $\theta$ phase behavior with mdPul input synapse timescales of $\tau_{rise} = 2ms$ and $\tau_{decay} = 10ms$, the default values used throughout this paper. Left: input synapse opening variable of a 1s-long simulation. Middle: Raster plot of a 1s-long simulation. Right: power spectrum of the LFP generated by the network in this 1s-long simulation. (**B**). LIP good $\theta$ phase behavior with shorter mdPul input synapse timescales of $\tau_{rise} = 0.1ms$ and $\tau_{decay} = 0.5ms$. Subplots are as described for A. (**C**). LIP good $\theta$ phase behavior with longer mdPul input synapse timescales of $\tau_{rise} = 10ms$ and $\tau_{decay} = 50ms$. Subplots are as described for A.

# Appendix 3

## Influence of Input Parameters on FEF Visuomotor Module Behavior

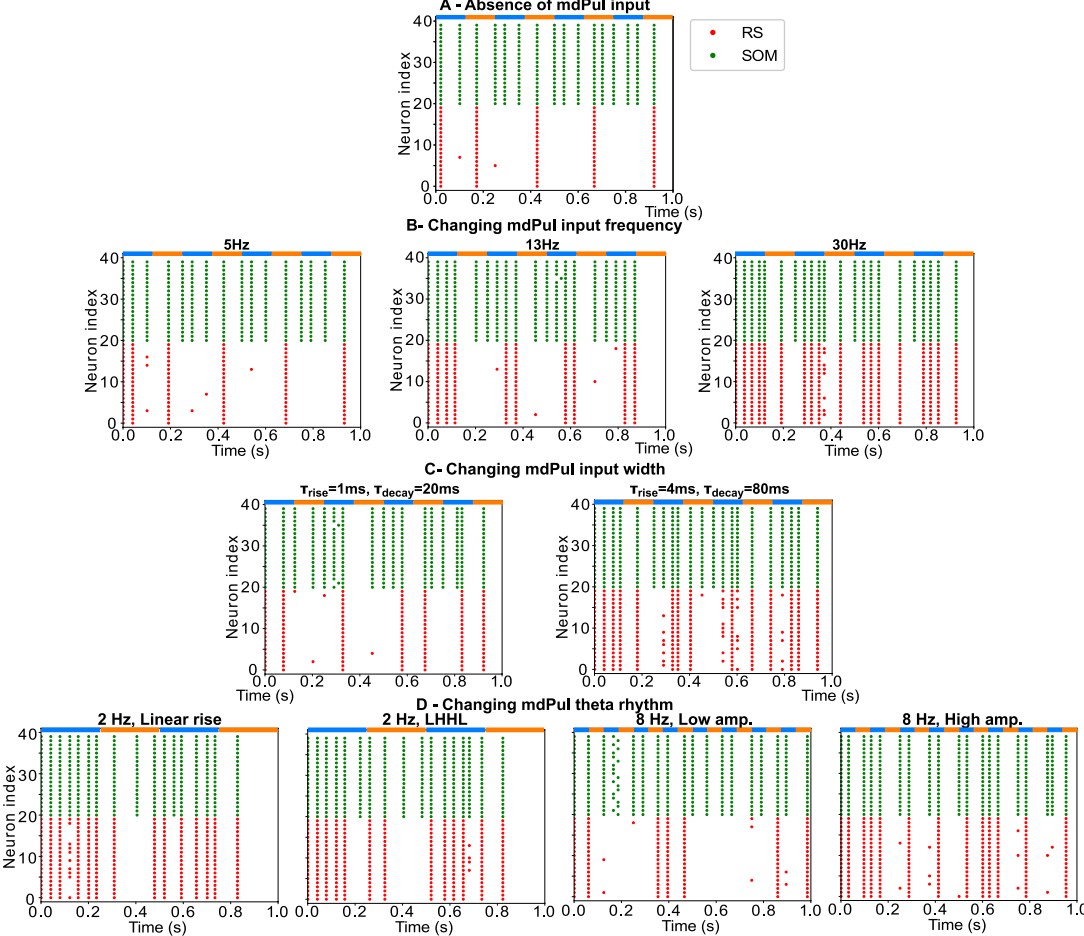

**Appendix 3—figure 1.** FEF visuomotor module good $\theta$ phase behavior is influenced by mdPul input properties. (**A**). Raster plot of FEF visuomotor behavior without mdPul input during a 1s-long simulation with alternating good and poor $\theta$ phase (no $\beta_2$ rhythm appears in the good $\theta$ phase). (**B**). Raster plots of FEF visuomotor behavior with various mdPul input frequencies, during a 1s-long simulations with alternating good and poor $\theta$ phase. Left: 5 Hz mdPul input (the $\beta_2$ rhythm is not sustained). Middle: 13 Hz mdPul input (the $\beta_2$ rhythm is preserved). Right: 30Hz mdPul input (the $\beta_2$ rhythm is speeds up). (**C**). Raster plot of FEF visuomotor good $\theta$ phase behavior with altered mdPul input synapse timescales. Left: $\tau_{rise} = 1ms$ and $\tau_{decay} = 20ms$ (the $\beta_2$ rhythm is not preserved). Right: $\tau_{rise} = 4ms$ and $\tau_{decay} = 80ms$ (the $\beta_2$ rhythm is not preserved). (**D**). Raster plots of FEF visuomotor behavior, during a 1s-long simulations with alternating good and poor $\theta$ phase with various frequencies. Left: 2Hz " $\theta$ " input, with linearly increasing maximal conductance, and with increasing then decreasing maximal conductance (the good $\theta$ phase $\beta_2$ is mostly preserved, with spikes appearing during the poor $\theta$ phase). Right: 8Hz " $\theta$ " input with the same maximal conductance as the low and high amplitude 4 Hz pulses.

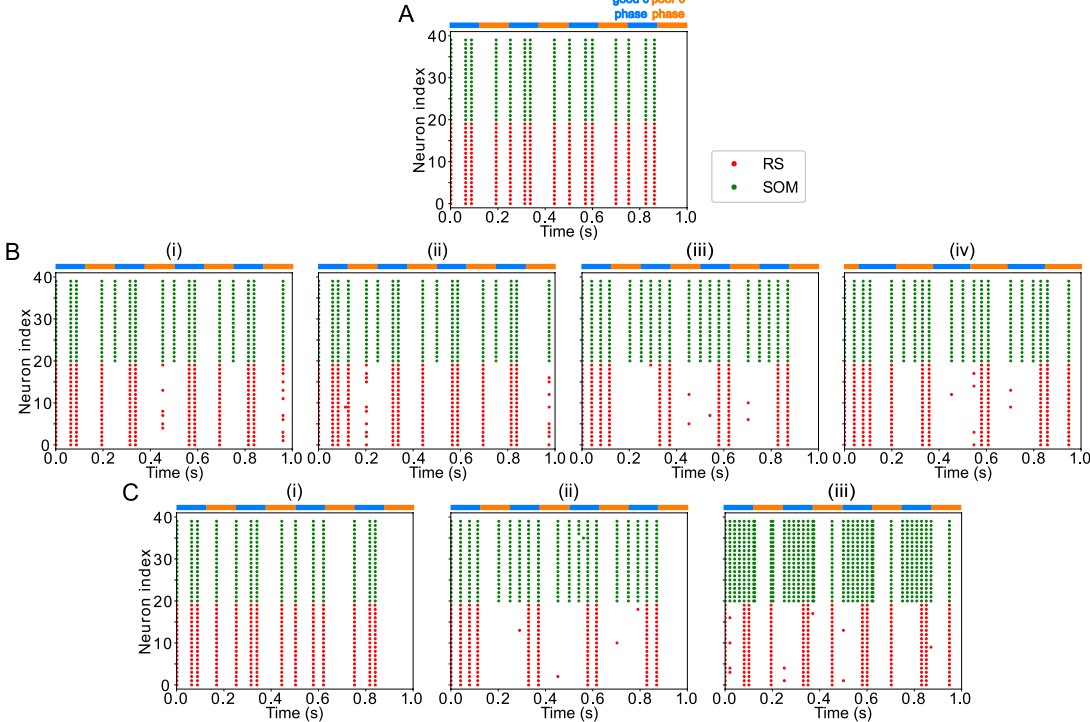

**Appendix 3—figure 2.** FEF visuomotor module good $\theta$ phase behavior is influenced by LIP input properties. (**A**). Raster plot of FEF visuomotor behavior without LIP input during a 1s-long simulation with alternating good and poor $\theta$ phase (the good $\theta$ phase $\beta_2$ rhythm is altered and spikes appear in the poor $\theta$ phase). (**B**). Raster plots of FEF visuomotor behavior with various LIP input frequencies during 1s-long simulations with alternating good and poor $\theta$ phase. (**i**) 5Hz LIP input speeds the good $\theta$ phase $\beta_2$ rhythm. (**ii**) 13Hz LIP input speeds the good $\theta$ phase $\beta_2$ rhythm. (**iii**) 25Hz LIP input preserves the good $\theta$ phase $\beta_2$ rhythm. (**iv**) 150Hz LIP input speeds the good $\theta$ phase $\beta_2$ rhythm. (**C**). Raster plots of FEF visuomotor behavior with various LIP input strengths during 1s-long simulations with alternating good and poor $\theta$ phase. (**i**) $1mS/cm^2$ input. (**ii**) $5mS/cm^2$ input. (**iii**) $15mS/cm^2$ input.

Our explorations indicate that a broad range of tonic excitation levels to FEF visuomotor RS cells result in quiescence during the poor θ phase and rhythmicity in RS and SOM cells during the good θ phase (S8). Too much tonic excitation to RS cells compared to SOM cells (*Appendix 3—figure 3*) results in RS cell firing during the poor θ phase despite SOM cell inhibition, while with too little tonic excitation to RS cells compared to SOM cells, RS cells are unable to take control of SOM cells outside of the good θ phase (when mdPul inputs provide windows of excitation), resulting in θ-rhythmic firing.

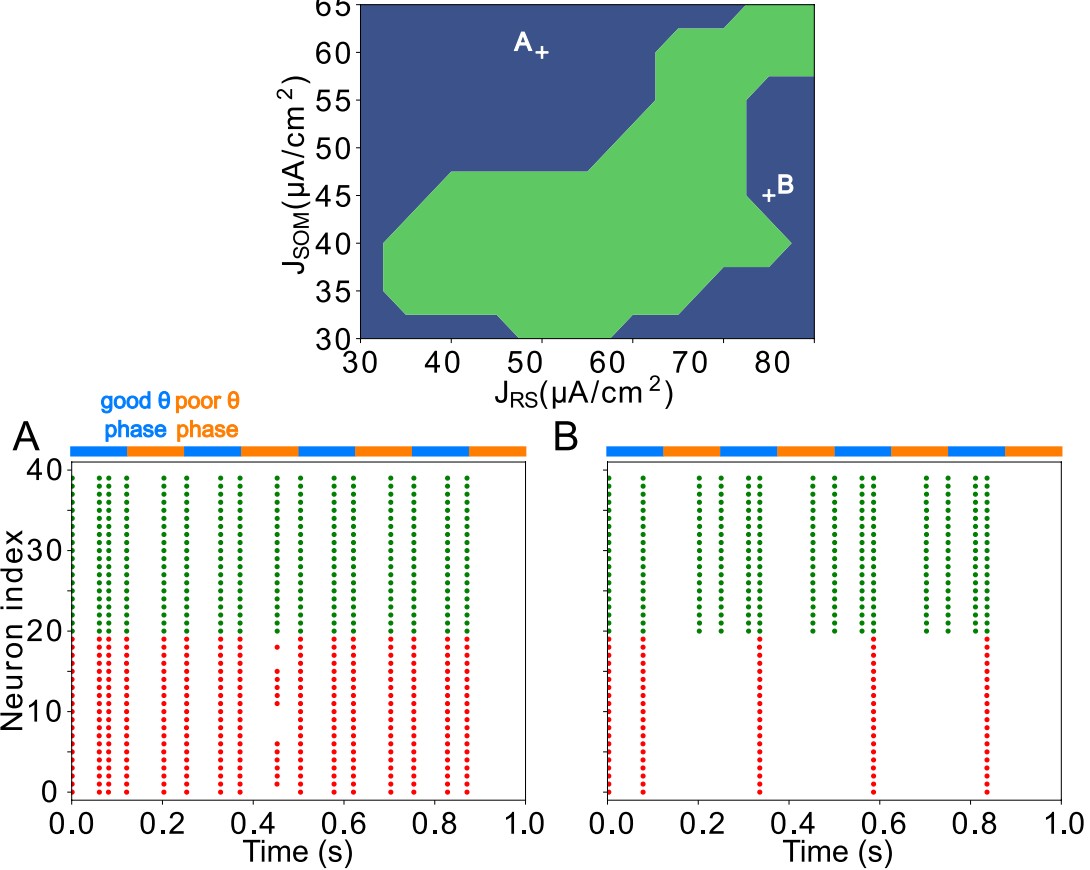

**Appendix 3—figure 3.** FEF visuomotor module good and poor $\theta$ phase behavior depends on the tonic excitation given to RS and SOM cells Top: Appropriate tonic excitation range needed to obtain a $\beta_2$ rhythm in the good $\theta$ phase and quiet RS cells in the poor $\theta$ phase. The crosses indicate the two sets of values used in simulations for A and B below. (**A**). Insufficient tonic excitation to SOM cells disturbs the poor $\theta$ phase by allowing RS cells spikes. (**B**). Excessive tonic excitation to SOM cells disturbs the good $\theta$ phase $\beta_2$ rhythm by silencing RS cells outside the windows of excitation provided by mdPul.

## Appendix 4

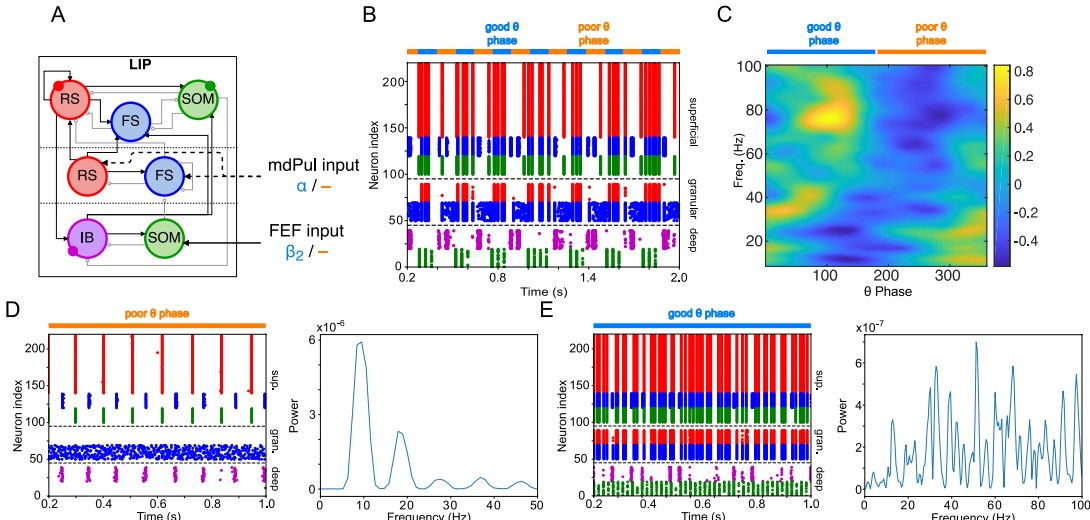

**Appendix 4—figure 1.** LIP module behavior persists with the addition of recurrent connections between RS cells. (**A**). Diagram of the LIP module with recurrent RS synapses. Recurrent synapses between RS cells have the same maximal conductance as RS→FS synapses, $1/40.mS/cm^2$ Symbols are as in *Figure 1C*. (**B**). LIP module spiking during a 1-second long simulation with synthetic pulvinar and FEF inputs. (**C**). Spectral power as a function of $\theta$ phase for the LIP module. Simulated LFP obtained from the sum of RS membrane voltages. See Methods. (**D**). Reproduction of LIP $\beta_1$ oscillatory rhythm, characteristic of the poor $\theta$ phase. Left, raster plot of a 1-second simulation of LIP activity with no input, corresponding to the poor $\theta$ phase, showing a $\beta_1$ oscillatory rhythm obtained through period concatenation. A single cycle of this rhythm is labeled: 1, IB cells spike by rebound; 2, superficial FS cells spike in response to IB excitation; 3, superficial RS cells spike by rebound; 4, superficial SOM cells spike in response to RS excitation. Right, power spectrum of the LFP generated by superficial RS cells for the same simulation, with a peak in the $\beta_1$ frequency band. (**E**). Reproduction of LIP γ oscillatory rhythm, characteristic of the good $\theta$ phase. Left, raster plot of a 1-second simulation of LIP activity in the presence of inputs, corresponding to the good $\theta$ phase, showing γ rhythms in the granular and superficial layers. Right, power spectrum of the LFP generated by superficial RS cells for the same simulation, with a peak in the γ frequency band.

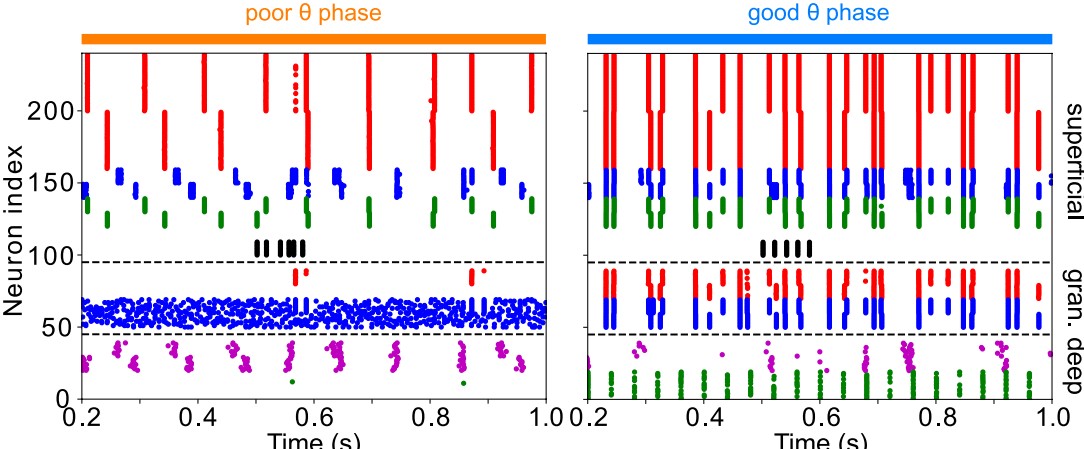

**Appendix 4—figure 2.** The addition of VIP interneurons to form a target-detection module in LIP superficial layer does not alter the network response to low-contrast targets. Simulations are for LIP superficial layer with the addition of VIP interneurons, connected to RS, FS, and SOM cells as in the FEF visual module, but receiving target input twice as strong. During the poor $\theta$ phase, target input delays firing of superficial RS cells; during the good $\theta$ phase, target input has no discernable effect on superficial RS cell firing.

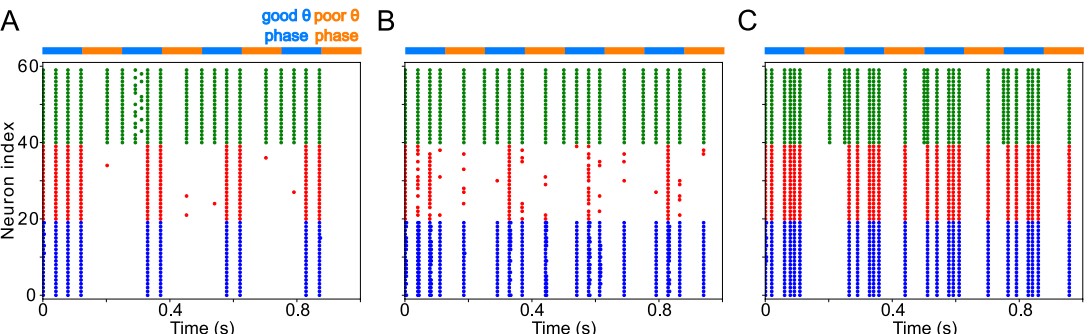

**Appendix 4—figure 3.** FEF visuomotor module behavior persists with the addition of an FS cell population, provided FS inhibition is not stronger than SOM inhibition. (**A**). Setting the conductance of synapses between the RS and FS population at the same level as synapses between RS and SOM cells preserves the $\beta_2$ rhythm in FEF. ($g_{RS->FS} = g_{RS->SOM} = 0.5mS/cm^2$, $g_{FS->RS} = g_{SOM->RS} = 0.8mS/cm^2$). (**B**). Increasing the conductance of the synapse between the RS and FS populations reduces RS spiking and abolishes the $\beta_2$ rhythm. ($g_{RS->FS} = 2mS/cm^2$, $g_{RS->SOM} = 0.5mS/cm^2$, $g_{FS->RS} = 2mS/cm^2$ , $g_{SOM->RS} = 0.8mS/cm^2$.) (**B**). Decreasing the conductance of the synapses from SOM to RS populations shifts the frequency of RS spiking to a $\gamma$ rhythm (arising through a typical PING mechanism). ($g_{RS->FS} = 0.5mS/cm^2$ , $g_{RS->SOM} = 0.5mS/cm^2, g_{FS->RS} = 0.8mS/cm^2 g_{SOM->RS} = 0.2mS/cm^2$.).

