## [Editor Report]

This valuable work introduces a detailed computational model to elucidate the underpinnings of experimentally observed coordinated rhythmic dynamics across brain regions. It provides a solid step towards understanding rhythmic attention. The work will be of interest to neuroscientists working on brain rhythms and attention from a cognitive, systems or computational perspective.

---

## [Decision Letter]

**Decision letter after peer review:**

Thank you for submitting your article "Interacting rhythms enhance sensitivity of target detection in a fronto-parietal computational model of visual attention" for consideration by *eLife*. Your article has been reviewed by 2 peer reviewers, and the evaluation has been overseen by a Reviewing Editor and Floris de Lange as the Senior Editor. The reviewers have opted to remain anonymous.

Essential revisions:

After discussion, the reviewers and editors agreed that the work has potential but is currently missing some key aspects. In terms of generalizability, a clear explanation of the key mechanisms of the model is missing. The work focuses more on reproducing the data rather than introducing mechanisms that might be used by the brain for other tasks. Main concerns, as detailed in the reviews below, include lack of justification of parameter choices (which often are given without explanation), and regarding spatial structure in the model reflecting the spatial structure of the task and physiology, which currently is very limited.

*Reviewer #1 (Recommendations for the authors):*

1) The authors should provide more detailed explanations for the choice of the model architecture. In this way, the reader can understand if the selected model features correspond to assumptions based on experimental literature or, instead, model predictions. Several features of the model are based on rodent data (e.g. Karnani et al., 2016 and Zhang et al. 2014 at lines 278). Consistently with this choice, the authors should then explain the justification for the self-inhibition of the SOM population in the visuo-motor module of FEF (Figure 2) given that these connections are virtually absent in the mouse visual cortex (Pfeffer et al., 2013 Nat. Neurosci.). Other key architectural choices which differ from the results in Pfeffer et al. (2013) and need to be clarified are: FS cells being absent and SOM not inhibiting VIP in the visuo-motor FEF module. Additionally, In mice, there is a high density of VIP cells in the superficial layers (Kim et al., 2017 Cell) so the authors should justify the absence of VIP cells in the superficial layers of LIP. Also, the authors should justify why the RS cells in the superficial layers of LIP are not self-exciting given the existence of strong recurrent excitatory connections (e.g. Cossell et al., 2015 Nature). Finally, the author should explain how the different parameters described in tables 1-10 were chosen (including whether they originate from measurements in the rodent or in the primate brain). More specifically, the authors should explain which of these parameters are based on the experimental literature and which of them are chosen to reproduce the oscillatory dynamics presented in the paper.

2) It seems that the model is not taking into consideration the spatial structure of the task, except for the FEF visual cells. As the author mention in line 38, the experimental literature shows that the rhythm of hit rates depends on whether the target is at the cued (8 Hz) or uncued (4 Hz) location. I would have expected the model to have working memory neurons to encode the position of the cue and then decision cells to use this information and produce hit rates that depend on the position of the target relative to the cue. Instead, the authors limit their discussion regarding changes in hit rate rhythms to a possible change in LIP to FEF connectivity without any mention of the spatial structure of the network. If possible, the authors should include a more clear spatial structure to the model. Otherwise, the authors should provide a more clear explanation of their choice.

3) The model is fairly complex and it will benefit the paper to increase the explanatory power of most of the figures. In particular, the authors should add spectrograms of the RS cells described in Figure 3B to show how the different bands wax and wane as a function of the theta phase (this is shown only for the FEF visuomotor cells in Figure 5 at the moment). For example, it is not clear from Figure 3B that during the poor θ phase, LIP produces a β1 rhythm. Also, the authors should label all the cell types in the model outlines of Figure 3A, 4A, 5A and 6A, 10 and 11 and, possibly, mention in these panels which rhythm is present in each external input. Finally, I recommend moving Figure 2 to Figure 1 to facilitate a comprehensive understanding of the relationship between the model and the empirical findings.

*Reviewer #2 (Recommendations for the authors):*

1, Apparently, from the model setting, gap junctions play an essential role in getting the results. However, is there any experimental evidence showing that such widespread and strong gap junctional coupling (the modeling used in this paper) exists between LIP and FEF neurons? To me, there is very few reports of gap junction between these two brain areas. Although, generally speaking, gap junction has higher probability to appear between cortical interneurons, but the coupling coefficients (cc) of such gap junctional coupling is very small, compared to what used in this study. So, please justify why consider gap junctions between excitatory neurons, and why with such strong strengths? It appears that the gap junction used in the present paper is too strong, such that the spikings of LIP and FEF neurons are too synchronized, making the rhythms generated in these two areas appear to be rather "artificial" (see more questions see below). BTW, what does the "LIP superficial SI cells" refer to in line 711? This kind of cell is not reported above.

2, The synapse types (inhibitory / excitatory) between neurons are unclear in the model diagram of Figure 2, Figure 3A, Figure 4A, Figure 5(a) and Figure 6(a).

3, The authors stated their model is capable of producing the γ,β_1,β_2 rhythms in LIP and FEF modules, and explained the excitation and inhibition relationship within each module during the rhythm activity. However, it remains unclear why each module produces the oscillation activities at such specific oscillation bands, which should be the key issue of the working mechanism from the computational aspect. I doubt that by using too strong gap junctions and highly synchronized spiking inputs (from mdPul and V4), neuron activities are too easy to be synchronized, preventing us to inspect the true underlying mechanism for the rhythm generation. With such over-synchronized activities, the oscillation band produced in the model becomes trivially relying on the oscillation of the external input.

[Editors' note: further revisions were suggested prior to acceptance, as described below.]

Thank you for resubmitting your work entitled "Interacting rhythms enhance sensitivity of target detection in a fronto-parietal computational model of visual attention" for further consideration by *eLife*. Your revised article has been evaluated by Floris de Lange (Senior Editor), a Reviewing Editor, and one of the original reviewers.

The manuscript has been improved but there are some remaining issues that need to be addressed, as outlined below.

*Reviewer #1 (Recommendations for the authors):*

I am glad to receive this revised version of the manuscript. Its clarity has greatly improved, particularly that of the model's scope and choices. The authors addressed thoroughly my comments. I believe the current manuscript is stronger and more compelling. However, there are some remaining points listed below that I believe must be addressed.

1) Excluding biological elements that are deemed not necessary is a parsimonious and helpful approach. However, there are cases in which the manuscript is not sufficiently convincing that these elements are not necessary. For example, the different cell types form a complex microcircuit. Removing an element from this microcircuit could break down its normal functioning. The reverse of this concept is the following: the authors show that modules, where specific cell types have been removed, mimic the neural dynamics observed experimentally. However, they often do not prove that putting back this cell type, which likely exists in this area, will not generate dynamics inconsistent with the data. This question is addressed at least once by the authors in line 771:

"in our hands, placing a target-detection circuit in LIP superficial layers -like the one modeled in the FEF module-did not alter LIP rhythms or improve detection of low contrast targets."

It is great that the author did this analysis but then they should add a reference here to the related figure (it is not clear which one). Similarly, the authors should test, not necessarily in the full model, but at least in the independent modules, what would happen if they add the FS neurons to the visuomotor FEF module or the recurrent RS connections in the LIP module and the visual FEF module. I can imagine two possibilities: (i) nothing changes qualitatively, in which case the authors' approach is justified because it uses a more parsimonious model without unnecessary elements; (ii) the behavior is very different for the conditions tested, in which case the authors should comment on that.

---

## [Author Response]

Essential revisions:After discussion, the reviewers and editors agreed that the work has potential but is currently missing some key aspects. In terms of generalizability, a clear explanation of the key mechanisms of the model is missing. The work focuses more on reproducing the data rather than introducing mechanisms that might be used by the brain for other tasks. Main concerns, as detailed in the reviews below, include lack of justification of parameter choices (which often are given without explanation), and regarding spatial structure in the model reflecting the spatial structure of the task and physiology, which currently is very limited.

We thank the reviewers and editors for their careful reading of our manuscript, and their detailed and insightful comments and questions. In the revised manuscript, we more clearly explain (i) our modeling strategy, (ii) the scope of the current model, (iii) the modeling choices we have made (including parameter choices), and (iv) the model’s mechanistic underpinnings.

In particular, we have clarified that the current model is not a model of the full Egly-Driver task and its spatial structure. Rather, it is a model of how *θ*-rhythmic input from the mediodorsal pulvinar (mdPul) to cortex, subsequent to a spatially informative cue, can produce alternating dynamical regimes that also explain periodicity in behavioral outcomes (i.e., hit rates). Our goal is thus to explain how periodic mdPul input produces the observed cortical dynamics, and how the observed cortical dynamics result in the observed behavioral patterns. As a result, we have omitted some details that may be important for the full task but are less relevant for the simplified (but hardly simple) questions that we address with the current model. We have also clarified how the choice of a relatively small model constrains the realism of our simulations, e.g., resulting in more regular rhythms than those observed in vivo.

In addition to clarifying these points, we have corrected the discrepancy pointed out by Reviewer #1 between the experimental data reported in Pfeffer *et al.* and the selfinhibition between our model FEF visuo-motor SOM cells. The FEF visuo-motor module is now simplified, with mdPul input exciting RS cells directly, and mdPul-driven RS cell activity sculpted by recurrent SOM cell inhibition to produce a β_2_ rhythm. We have also expanded our discussion of how changing frequency (via the inhibitory time constants of SOM and FS interneurons) alters task performance. And we have added several new plots and thoroughly revised the figures to make them more legible and informative. These changes include labeling the cell types and indicating the frequencies of input rhythms in the diagrams, altering the colors of diagrams and plots to be more distinguishable and coherent, and increasing the font sizes throughout. We hope you will agree that this revised manuscript is clearer, more complete, more generalizable, and worthy of publication in *eLife*.

Reviewer #1 (Recommendations for the authors):1) The authors should provide more detailed explanations for the choice of the model architecture. In this way, the reader can understand if the selected model features correspond to assumptions based on experimental literature or, instead, model predictions. Several features of the model are based on rodent data (e.g. Karnani et al., 2016 and Zhang et al. 2014 at lines 278). Consistently with this choice, the authors should then explain the justification for the self-inhibition of the SOM population in the visuo-motor module of FEF (Figure 2) given that these connections are virtually absent in the mouse visual cortex (Pfeffer et al., 2013 Nat. Neurosci.). Other key architectural choices which differ from the results in Pfeffer et al. (2013) and need to be clarified are: FS cells being absent and SOM not inhibiting VIP in the visuo-motor FEF module. Additionally, In mice, there is a high density of VIP cells in the superficial layers (Kim et al., 2017 Cell) so the authors should justify the absence of VIP cells in the superficial layers of LIP. Also, the authors should justify why the RS cells in the superficial layers of LIP are not self-exciting given the existence of strong recurrent excitatory connections (e.g. Cossell et al., 2015 Nature). Finally, the author should explain how the different parameters described in tables 1-10 were chosen (including whether they originate from measurements in the rodent or in the primate brain). More specifically, the authors should explain which of these parameters are based on the experimental literature and which of them are chosen to reproduce the oscillatory dynamics presented in the paper.

We appreciate the reviewer’s detailed comments comparing our model’s architecture to experimental data. In particular, we thank the reviewer for pointing out the discrepancy between our use of self-inhibition between FEF visuo-motor SOM cells and the experimental data reported in Pfeffer et al. For the revised manuscript, we have addressed this issue by simplifying the FEF VM module, which no longer includes SOM self-inhibition or VIP cell disinhibition. In the new module, mdPul input excites RS cells directly, rather than disinhibiting RS cells via activation of VIP cell-mediated inhibition of SOM cells. mdPul-driven RS cell activity is then sculpted by recurrent SOM cell inhibition to produce a β_2_ rhythm.

The reviewer will notice that in the new FEF VM module, we omit not only the FS cell population but also the VIP cell population. This absence, as well as the absence of VIP cells and self-excitation among superficial RS cells in LIP (as pointed out by the reviewer), are a consequence of our particular modeling philosophy in the context of this work. Since it is impossible to include all the details of brain physiology and anatomy, a part of the modeling process is to choose those biological features we believe are most relevant to the task we are modeling. In particular, the spatial structure of the full EglyDriver task is beyond the scope of the current model. The full task, involving spatially informative cues and spatially selective subpopulations responding to them, requires the regulation of which subpopulations in each module are activate at a given time. This greater mechanistic complexity, in addition to involving a detailed pulvinar model and inputs from other regions, may also involve VIP cells and self-excitation between RS cells. VIP cells may participate, e.g., by mediating the disinhibition of spatially selective populations. Excitatory synapses among cortical pyramidal cells, which are extremely important for plasticity and formation of cell assemblies, may also be important in the selection of spatially specific subpopulations. But neither are necessary for the current model (a stepping stone to a more complex model of the full Egly-Driver task), in which all the cells in LIP can be thought of as part of a single, cue-relevant cell assembly. As explained in the manuscript, we include VIP cells in the FEF visual module because the division of FEF visual cells into multiple populations enhances target detection. (In fact, in an earlier version of the model, we attempted to place a version of the target detection module in superficial LIP, and found that it did not enhance the detection of low-contrast targets.) In the revised manuscript, we have included text very much like the preceding paragraph in the Discussion, starting on line 741, and a further discussion of VIP cells in superficial LIP on lines 782–788 (excerpted below in response to point 10). We have also attempted to explain our modeling strategy more clearly in the Introduction, which starting at line 97 now reads:

“The model was constructed from biologically detailed neurons and is consistent with the known anatomy, physiology, and function of frontal and parietal circuits. It highlights the cell types and connections necessary to produce the dynamics of interest, to allow realistic insights into the mechanisms underlying these dynamics and their functional consequences, while also avoiding excessive complexity. That is, while other cell types and connections, besides those modeled, are present in the brain and may play important roles in other tasks and other aspects of this task, the present model suggests that the activity of these cell types plays a minimal role in the phenomena of interest (see Discussion, Modeling Choices and Limitations).”

We also thank the reviewer for prompting us to clearly specify the origins of the parameters used in the model. We have attempted to do so in the revised Methods section—see lines 867, 873, 878, 883, and 887–892. Regarding our parameter choices, in the revised manuscript, the paragraph above continues:

“Since small models with single-compartment neurons can be difficult to parameterize using electrophysiological measurements, we built on previous cortical models that reproduced experimentally observed network dynamics in other cognitive and physiological contexts—specifically, in the context of working memory and cholinergic neuromodulation (Kramer et al., 2008; Lee et al., 2013; Gelastopoulos et al., 2019). Incorporating these models as modules in our network increases the realism and explanatory power of our model, by further constraining its dynamics and function to match experimental observations.”

2) It seems that the model is not taking into consideration the spatial structure of the task, except for the FEF visual cells. As the author mention in line 38, the experimental literature shows that the rhythm of hit rates depends on whether the target is at the cued (8 Hz) or uncued (4 Hz) location. I would have expected the model to have working memory neurons to encode the position of the cue and then decision cells to use this information and produce hit rates that depend on the position of the target relative to the cue. Instead, the authors limit their discussion regarding changes in hit rate rhythms to a possible change in LIP to FEF connectivity without any mention of the spatial structure of the network. If possible, the authors should include a more clear spatial structure to the model. Otherwise, the authors should provide a more clear explanation of their choice.

We thank the reviewer for these comments, which provide another opportunity to clarify our strategy and intentions for the current model. We believe that spatial structure is very important to explain the Egly-Driver task, which is a spatial attention task. However, we feel that the mechanisms of spatial selection are of sufficient complexity as to be beyond the scope of the current modeling project, which is already quite complex. The question we address in the current manuscript is how the observed dynamics of visual processing at a single visual location (or object) produce alternating periods of enhanced and diminished perceptual sensitivity. In ongoing work, we are addressing how further brain regions, including the pulvinar and the superior colliculus, coordinate these temporal dynamics between locations. We believe that it is these regions that encode the cue location and alter the decision criteria for target detection. The current model revealed a previously unexpected mechanism by which the hit rate rhythm might be altered at a single location, i.e., by changing LIP to FEF connectivity. Differences in LIP to FEF connectivity among neural populations representing different locations in the Egly-Driver task might explain previously observed differences in the frequency of enhanced visual processing. Representations of spatial information and the statistics of target presentation (possibly in brain regions such as mdPul and the superior colliculus) may alter LIP to FEF connectivity, resulting in different frequencies of visual sampling at the cued and uncued locations.

This clarifying information has been added to the Introduction and Discussion of the revised manuscript. In the Introduction, after listing our questions of interest starting on line 85, the manuscript now reads (starting at line 91):

“To address these questions, we built and tested a computational model of FEF and LIP, representing the temporal dynamics of visual processing at a single location (on a single object), following the presentation of an attention-priming spatial cue. We leave the allocation of attention among multiple objects for future work. In our model, FEF and LIP were both driven θ-rhythmically by simulated mdPul input at α frequency. This is consistent with the experimental observation that mdPul exerts a Granger causal influence over FEF and LIP at an α frequency during a part of each θ cycle (i.e., at a particular phase of the θ rhythm; Fiebelkorn et al. (2013, 2019)).”

The Discussion, starting at line 735, now reads:

“A deeper understanding of the role of the θ rhythm awaits a detailed investigation of pulvinar activity and the mechanisms of θ rhythmicity and spatial selection in this task. In ongoing work, we are extending the current model to explore how circuits in the pulvinar and the superior colliculus may coordinate ongoing periodic temporal dynamics, as modeled here, among multiple identical networks representing different spatial locations and/or objects.”

3) The model is fairly complex and it will benefit the paper to increase the explanatory power of most of the figures. In particular, the authors should add spectrograms of the RS cells described in Figure 3B to show how the different bands wax and wane as a function of the theta phase (this is shown only for the FEF visuomotor cells in Figure 5 at the moment). For example, it is not clear from Figure 3B that during the poor θ phase, LIP produces a β1 rhythm. Also, the authors should label all the cell types in the model outlines of Figure 3A, 4A, 5A and 6A, 10 and 11 and, possibly, mention in these panels which rhythm is present in each external input. Finally, I recommend moving Figure 2 to Figure 1 to facilitate a comprehensive understanding of the relationship between the model and the empirical findings.

We thank the reviewer for these helpful suggestions. We have labeled the cell types in the model schematics, and indicated the frequency of input rhythms. We have consolidated Figure 2 and Figure 1, as suggested. In the experimental data (*Fiebelkorn et al. 2019*), power is plotted as a function of theta phase; to match this result, we have plotted power as a function of theta phase in Figure 3B (formerly Figure 4B) and in Figure 4 (formerly Figure 5).

Reviewer #2 (Recommendations for the authors):1, Apparently, from the model setting, gap junctions play an essential role in getting the results. However, is there any experimental evidence showing that such widespread and strong gap junctional coupling (the modeling used in this paper) exists between LIP and FEF neurons? To me, there is very few reports of gap junction between these two brain areas. Although, generally speaking, gap junction has higher probability to appear between cortical interneurons, but the coupling coefficients (cc) of such gap junctional coupling is very small, compared to what used in this study. So, please justify why consider gap junctions between excitatory neurons, and why with such strong strengths? It appears that the gap junction used in the present paper is too strong, such that the spikings of LIP and FEF neurons are too synchronized, making the rhythms generated in these two areas appear to be rather "artificial" (see more questions see below).

We apologize for any lack of clarity on our part: the synapses from LIP to FEF are chemical, not electrical. We have done our best to clarify this issue in the text. We now specify on line 372:

“We varied the functional connectivity from LIP to the FEF visual module by altering the strength of the chemical synapses from superficial LIP RS cells to FEF visual cells.”

We mention that the synapses are chemical again on lines 379, 586, and 601, and in the legend of Figure 7.

There are three LIP cell populations that we do model with gap junctional connectivity (which is modeled within and not between populations) – superficial RS cells, superficial SOM cells, and IB cells. For these cell types, there is ample experimental evidence of the existence of electrical synapses. We understand the reviewer’s perception that the gap junction conductances between RS and SOM cells appear unusually high. In particular, the conductance between SOM cells is two orders of magnitude greater than that between IB axons. However, a number of studies suggest that SOM cell networks are extensively connected by gap junctions, and that electrical coupling plays a large role in determining the impact of these networks on pyramidal cells (Karnani *et al.* 2016; Safari *et al.* 2017) and their dynamics of these networks, including the generation of rhythmicity at θ frequencies (Beierlein, Gibson, and Connors 2000; Fanselow, Richardson, and Connors 2008; Hu and Agmon 2015; Huang *et al.* 2020). We believe these results justify the importance of these gap junctions for our model. This information is included in the revised manuscript, beginning on line 745 (excerpted below in our response to this reviewer’s point 4).

Regarding the realism of the dynamics in our model, as we describe above and in the Introduction of the revised manuscript, our goal is to allow realistic insights into the mechanisms and functional consequences of brain dynamics while prioritizing clarity and avoiding excessive complexity. To reach this goal, we have found it effective to use relatively small networks with model neurons that exhibit simplified but biophysically realistic dynamics and timescales. In order for the behavior of such small networks to qualitatively match the dynamics of much larger networks, their model parameters often must be different from those of the larger networks. For example, because our model contains only 80 RS cells, 20 SOM cells, and 20 IB cells, we cannot represent sparsely connected electrically coupled networks which involve a large number of cells, nor can we represent the impact of many gap junctions with a low conductance impinging on each cell in the network. To compensate for the small size of our network, we connect cell populations with all-to-all gap junctions, and adjust the conductances of these gap junctions to achieve a qualitative match for observed network dynamics (such normalization is standard in the modeling literature). To reach the goal of clarity with minimal complexity, we also accept rhythms that may be slightly “artificial” in ways that we believe are irrelevant to understanding the qualitative dynamics of the network and the functional implications we are exploring. For example, rhythms that are more regular than those seen in vivo. We do not believe that our model is dominated by gap junctional connectivity in any way that distorts the dynamics and function of the network. Of course, we could be wrong. This could be shown by examining a much larger model, but we believe that generating the hypotheses about model dynamics necessary for such tests relies on initial modeling studies in smaller, simplified networks. We have now explained this in the Discussion and Methods sections of the revised manuscript. On line 922, the Methods section now reads:

“Of note, the maximal conductances of these gap junctions were likely larger than what would be measured for single pairs of cells in vivo, as is appropriate for currents in a smaller network.”

Beginning on line 741, the Discussion now reads:

“As described in the Introduction, our model represents a number of simplifications relative to the full dynamics of the visual attention network in the context of this task. The model presented here is intended to allow realistic insights into the mechanisms underlying observed brain dynamics and their functional consequences while avoiding excessive complexity. Thus, we have chosen to highlight the cell types and connections necessary to produce the dynamics of interest, while leaving out other cell types and connections. …We have also chosen to model relatively small networks of biophysically realistic neurons. In order for the behavior of small networks to qualitatively match the dynamics of much larger networks, their model parameters often must be different from those of the larger networks. Their dynamics also may be different, in ways that we believe are irrelevant to understanding the qualitative dynamics of the network and the functional implications of interest – for example, rhythms that are more regular than those seen in vivo. Testing the conclusions of models such as ours in larger, more detailed models, as well as in experiments, is essential. However, initial modeling studies in smaller, simplified networks are indispensable in generating initial hypotheses about the mechanisms and functions of brain dynamics.”

BTW, what does the "LIP superficial SI cells" refer to in line 711? This kind of cell is not reported above.

We used the term “SI cells” to describe SOM cells in a previous version of the manuscript; this error has been corrected in the revised manuscript.

2, The synapse types (inhibitory / excitatory) between neurons are unclear in the model diagram of Figure 2, Figure 3A, Figure 4A, Figure 5(a) and Figure 6(a).

We have attempted to make the synapse types clearer, by using different colors and symbols for inhibitory (black circle) and excitatory (gray triangle) synapses.

3, The authors stated their model is capable of producing the γ,β_1,β_2 rhythms in LIP and FEF modules, and explained the excitation and inhibition relationship within each module during the rhythm activity. However, it remains unclear why each module produces the oscillation activities at such specific oscillation bands, which should be the key issue of the working mechanism from the computational aspect. I doubt that by using too strong gap junctions and highly synchronized spiking inputs (from mdPul and V4), neuron activities are too easy to be synchronized, preventing us to inspect the true underlying mechanism for the rhythm generation. With such over-synchronized activities, the oscillation band produced in the model becomes trivially relying on the oscillation of the external input.

We thank the reviewer for raising this important point. In the revised manuscript, we more completely explain how the intrinsic currents and timescales of each module give rise to oscillations at particular frequencies. We have addressed the role of gap junctions above, and we believe the synchrony of spiking inputs also does not overly constrain the dynamics of our model. In particular, the input rhythms are at different frequencies (α and θ) than the rhythms explained by our model (γ and β). This makes it unlikely that the rhythms produced in our model rely trivially on the input rhythms.

[Editors' note: further revisions were suggested prior to acceptance, as described below.]

The manuscript has been improved but there are some remaining issues that need to be addressed, as outlined below.

We thank the reviewer for pointing out these issues, which we have addressed below and in the revised manuscript.

We have also further corrected the model by removing recurrent SOM synapses in LIP, following a comment made by the reviewer in the previous round of reviews with regards to such connections in FEF. This change to the model has resulted in some changes in the figures and text, but our overall conclusions remain valid. In particular, we have discovered that the 8 Hz rhythmicity in hit rate is dependent in part on the shape of mdPul input. Additionally, given a clearer understanding of model behavior, we have reframed our simulations as primarily modeling the neurophysiology and behavior of targets presented at the cued location.

We have also made several other small improvements to the manuscript and figures. We have increased the duration of some raster plots to 2 seconds to better illustrate model behavior in Figures 2, 3, and 4, and we have labeled the good and poor θ phases in all raster plots. Finally, we have expanded the discussion to make connections with a broader literature.

We hope you will agree that the manuscript is improved and ready for publication.

Reviewer #1 (Recommendations for the authors):I am glad to receive this revised version of the manuscript. Its clarity has greatly improved, particularly that of the model's scope and choices. The authors addressed thoroughly my comments. I believe the current manuscript is stronger and more compelling. However, there are some remaining points listed below that I believe must be addressed.

We thank the reviewer for the kind words, and for another close and critical reading of our manuscript. We have addressed the reviewer’s comments and taken most of their suggestions in the revised manuscript.

1) Excluding biological elements that are deemed not necessary is a parsimonious and helpful approach. However, there are cases in which the manuscript is not sufficiently convincing that these elements are not necessary. For example, the different cell types form a complex microcircuit. Removing an element from this microcircuit could break down its normal functioning. The reverse of this concept is the following: the authors show that modules, where specific cell types have been removed, mimic the neural dynamics observed experimentally. However, they often do not prove that putting back this cell type, which likely exists in this area, will not generate dynamics inconsistent with the data. This question is addressed at least once by the authors in line 771:"in our hands, placing a target-detection circuit in LIP superficial layers -like the one modeled in the FEF module-did not alter LIP rhythms or improve detection of low contrast targets."It is great that the author did this analysis but then they should add a reference here to the related figure (it is not clear which one). Similarly, the authors should test, not necessarily in the full model, but at least in the independent modules, what would happen if they add the FS neurons to the visuomotor FEF module or the recurrent RS connections in the LIP module and the visual FEF module. I can imagine two possibilities: (i) nothing changes qualitatively, in which case the authors' approach is justified because it uses a more parsimonious model without unnecessary elements; (ii) the behavior is very different for the conditions tested, in which case the authors should comment on that.

In response to the author’s suggestion, we have added supplementary figures showing that three different modified modules exhibit qualitatively similar dynamics to our original model:

An LIP module exhibiting recurrent RS excitatory synapses.An LIP module exhibiting a target-detection circuit in the superficial layers driven by granular layer inputs (including a target).An FEF visuomotor module containing an FS cell population.

These appear in Figures S9, S10, and S11, respectively, which are cited in the text at line 822 (to which the reviewer refers above) as well as at line 788:

“The qualitative dynamics of the LIP module remained the same when recurrent connections between RS cells were added (Figure S9) and when VIP cells were added to the superficial layer (as part of a target detection module like the one instantiated in the FEF visual module; Figure S10). Similarly, the addition of FS cells in the FEF visuomotor module also did not significantly alter this module's β2 dynamics (Figure S11).”